# PyramidCLIP: Hierarchical Feature Alignment for Vision-language Model Pretraining

**Yuting Gao**[1*]   **Jinfeng Liu**[1,2,*]   **Zihan Xu**[1,*]   **Jun Zhang**[1]   **Ke Li**[1]   **Chunhua Shen**[3]

[1]Tencent Youtu Lab    [2]Shanghai Jiaotong University    [3]Zhejiang University
{yutinggao,ianxxu,bobbyjzhang,tristanli}@tencent.com

## Abstract

Large-scale vision-language pre-training has achieved promising results on downstream tasks. Existing methods highly rely on the assumption that the image-text pairs crawled from the Internet are in perfect one-to-one correspondence. However, in real scenarios, this assumption can be difficult to hold: the text description, obtained by crawling the affiliated metadata of the image, often suffers from the *semantic mismatch* and the *mutual compatibility*. To address these issues, we introduce PyramidCLIP, which constructs an input pyramid with different semantic levels for each modality, and aligns visual elements and linguistic elements in the form of hierarchy via *peer-level semantics alignment* and *cross-level relation alignment*. Furthermore, we soften the loss of negative samples (unpaired samples) so as to weaken the strict constraint during the pre-training stage, thus mitigating the risk of forcing the model to distinguish compatible negative pairs. Experiments on five downstream tasks demonstrate the effectiveness of the proposed Pyramid-CLIP. In particular, with the same amount of 15 million pre-training image-text pairs, PyramidCLIP exceeds CLIP on ImageNet zero-shot classification top-1 accuracy by 10.6%/13.2%/10.0% with ResNet50/ViT-B32/ViT-B16 based image encoder respectively. When scaling to larger datasets, PyramidCLIP achieves the state-of-the-art results on several downstream tasks. In particular, the results of PyramidCLIP-ResNet50 trained on 143M image-text pairs surpass that of CLIP using 400M data on ImageNet zero-shot classification task, significantly improving the data efficiency of CLIP.

## 1 Introduction

Recently, vision-language pre-training (VLP) has achieved great success, which aims to improve the accuracy of downstream vision-language tasks by pre-training a model on large-scale image-text pairs harvested from the web without any manual annotation. The mainstream VLP methods can be categorized into two paradigms, single-stream (1; 2; 3; 4; 5) and dual-stream (6; 7; 8; 9; 10). Compared to the single-stream counterpart, the dual-stream paradigm decouples the image encoder and text encoder and extracts features for images and texts respectively. Due to its simplicity and flexibility for downstream applications, the dual-stream paradigm dominates. The representative dual-stream model CLIP (6) performs contrastive vision-language pre-training on 400M image-text pairs collected from the Internet, which achieves astounding results. Later, DeCLIP (10) and FILIP (8) improve CLIP by introducing more supervisions, and bringing in fine-grained cross-modal interaction.

Although existing CLIP-alike methods have achieved promising results on downstream tasks, they strongly rely on the assumption that the image-text pairs are of high quality, *i.e.*, the pairs are in

---

*The first three authors contributed equally. This work was done when J. Liu was an intern at Tencent.

perfect one-to-one correspondence and have no correlation with other unpaired samples. However, this ideal assumption is hard to satisfy in practice as shown in Figure 1. Firstly, semantic mismatch between the visual modality and linguistic modality exists, *e.g.*, (a) Caption Redundancy: the affiliated text description is redundant and contains irrelevant information; (b) Image Redundancy: the Region-of-Interest (ROI) corresponding to the text is only a sub-region of the image; and (c) Cast Deficiency: text misses the descriptions of main objects in the image, while visual modelling needs to reason about the relationship among salient instances. Secondly, captions might be compatible to some extent among pairs, as illustrated in Figure 1(d). Existing methods directly treat other pairs as negative samples regardless the potential correlation, which may confuse the model.

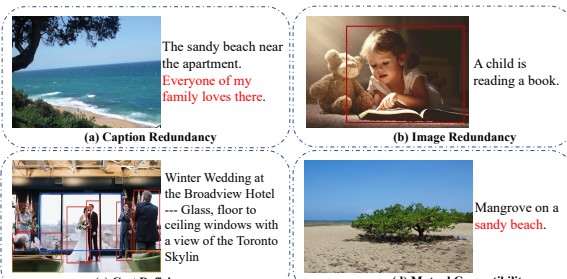

**Figure 1:** Problems in the web-crawled image-text pairs. (a)(b)(c) suffer the *semantic mismatch* between visual modality and linguistic modality, while (d) shows an example of the *mutual compatibility* with (a). Note that, in (a) the red caption is redundant; in (b) the image outside the red bounding box is the redundant; in (c) the descriptions for the casts in the red boxes are missing; and in (d) the red caption is compatible with the image of (a).

In order to tackle the issues mentioned above, we propose **PyramidCLIP** in this paper, which attempts to align image-text pairs more precisely in the form of hierarchy. PyramidCLIP constructs two input pyramids with different semantic levels at both sides of the dual-stream network, *i.e.*, the global image, local image region, and ROI features of the salient objects in the image for visual modelling; text summarization, the original caption and categories with attribute descriptions of salient objects for linguistic modelling. Then we contrast visual elements and linguistic elements via *peer-level semantics alignment* and *cross-level relation alignment*, tackling the issues of mentioned in (a), (b) and (c). Specifically, for peer-level semantics alignment, since the global image and text summarization both contain global semantic information, and the local region and original caption both contain more fine-grained semantic information, they are treated as two pairs of positive samples. For cross-level relation alignment, to avoid the modelling of object relationship being overwhelmed by the semantics modelling, we explicitly align the fine-grained object relation with cross-layer elements in another modality. Moreover, for the issue of the *mutual compatibility*, we soften the loss term of the negative unpaired samples during the contrast process to ease the strict constraint, alleviating the negative effect of unpaired similarities.

Extensive experiments demonstrate the effectiveness of our proposed PyramidCLIP. For fair comparison, when trained on YFCC15M-V2 (10) dataset, with ResNet50 (11)/ViT-B32 (12)/ViT-B16 (12) as the image encoder and Transformer as the text encoder, our model achieves the state-of-the-art (SoTA) zero-shot classification on ImageNet (13) with 47.8%/46.0%/50.7% top-1 accuracy. In comparison, the CLIP achieves 37.2%/32.8%/40.7% respectively. Furthermore, when scaling to the large-scale dataset, the results of PyramidCLIP achieve SoTA on several downstream tasks, in particular, the results of PyramidCLIP-ResNet50 trained on 143M image-text pairs surpass that of CLIP trained on 400M data on ImageNet zero-shot classification task, improving the data efficiency of CLIP significantly.

Our main contributions are summarized as follows: (*i*) We propose PyramidCLIP for more accurate image-text alignment for vision-language model pre-training, which effectively constructs two input pyramids at both sides of the visual encoder and linguistic encoder, and then align the visual elements and linguistic elements via peer-level semantics alignment and cross-level relation alignment. (*ii*) We soften the loss term of negative samples during the contrast process to ease the strict constraint, so as to alleviate the negative effect caused by local similarities. (*iii*) Extensive experiments demonstrate the effectiveness of PyramidCLIP, which achieves SoTA on several downstream tasks.

## 2   Related Work

**Vision-Language Pre-training** Vision-Language Pre-training (VLP) learns a strong joint representation between two modalities by pre-training models on large-scale image-text pairs. In terms of the model architecture, the mainstream VLP models can be divided into two types: single-stream and dual-stream. The former one uses a single transformer to model both image and text representations

in a unified semantic space by concatenating image and text input embeddings, including Visual-BERT (2), UNITER (1), UNICODER (4), OSCAR (3) and UNIMO (14). The latter one encodes images and texts separately with decoupled image encoder and text encoder, such as ViLBERT (15), LXMERT (9), ALIGN (7), CLIP (6), and DeCLIP (10). From a different perspective, the pre-training objective mainly comprises two categories: image-text contrastive learning and masked token tasks based on Language Modeling (LM). Among the methods mentioned above, UNIMO, ALIGN, CLIP and DeCLIP adopt contrastive learning to align the textual and visual representation in a unified semantic space. In contrast, VisualBERT, UNITER, LXMERT and UNICODER use masked token tasks, including Masked Language/Region Modeling and autoregressive LM. In this paper, we employ dual-stream architecture and contrastive learning for simplicity, flexibility, and relatively cheaper computation cost.

**Fine-grained Alignment** Due to the semantic gap between image and text, there may be some troubles in directly performing the alignment between these two modalities. For example, some words or phrases in the descriptions may be irrelevant to the images, or the corresponding descriptions of the objects in an image may not always be available in the caption. Thus, finer-grained alignment is indispensable, as it can provide more accurate and richer supervision signals of multiple granularity, improving the performance significantly. FILIP (8) improves the contrastive objective to achieve finer-level alignment by using a token-wise maximum similarity between visual and textual tokens. Methods in (3; 16; 17; 18) construct multi-level semantic concepts for finer-grained alignment. OSCAR (3) first introduces multi-level semantics, capturing object region features and the corresponding tags with a pre-trained object detector, then concatenates text, object tags and region features together to learn the joint representations. VinVL (16) enhances the visual representations of OSCAR by pre-trianing a more powerful object-attribute detector. Both OSCAR and VinVL form the multi-level semantics only in the visual modality. MVPTR (17) and X-VLM (18) obtain their multi-level semantics concepts in both visual and linguistic modalities. MVPTR limits the interaction between object tags and textual tokens, and learns the object-tag alignment in an explicit manner. It also models the nested property of language by learning phrase-level semantics. X-VLM learns multi-level alignments by positioning vision concepts using given texts, and makes alignments between these two parts. However, in addition to the image encoder and the text encoder, the two methods both have an additional cross-modal encoder, bringing computation overhead.

In this paper, we follow the dual-stream design of CLIP and construct three visual semantics levels and three linguistic semantics levels to form our PyramidCLIP. Different from methods mentioned above, each level is input to the corresponding encoder individually, without concatenating. The obtained three visual representations and three linguistic representations are used to compute six contrastive loss terms, which helps to achieve multi-level alignments.

## 3 Method

In this section, we introduce the proposed PyramidCLIP for more accurate alignment of image and text for vision-language model pre-training.

### 3.1 Overall Architecture

The entire framework of the proposed PyramidCLIP is presented in Figure 2. PyramidCLIP is a dual-stream network, including a text encoder $h$ and an image encoder $f = f_2 \circ f_1$, where $f_1$ and $f_2$ denote the front part and the rear part of the image encoder respectively. Each encoder consists of a linear projector and a normalization operator in the end, which project the final class token into a unified dimension and then normalize it, obtaining the corresponding visual or linguistic representation vector in the same embedded space.

During the training, for each image-text pair $(I, T)$, the image $I$ is transformed into two views, *i.e.*, local view $L$ and global view $G$, through random crop with different ratios. And text $T$ is input to a summary extractor (19) to generate text summarization $T_S$ with higher level semantics. The image global view $G$ and text summarization $T_S$ both capture global context information, while the image local view $L$ and original text $T$ contain more detailed information. Therefore, $G$ and $T_S$ are regarded as a pair of positive samples, while $L$ and $T$ are regarded as another pair of positive samples, denoted as $(G, T_S)$ and $(L, T)$. These two pairs are input to the dual-stream encoder to extract global and local representation pairs, $(\boldsymbol{v}^g, \boldsymbol{l}^s)$ and $(\boldsymbol{v}^l, \boldsymbol{l}^t)$, where $\boldsymbol{v}^g = f(G)$, $\boldsymbol{l}^s = h(T_S)$, $\boldsymbol{v}^l = f(L)$ and

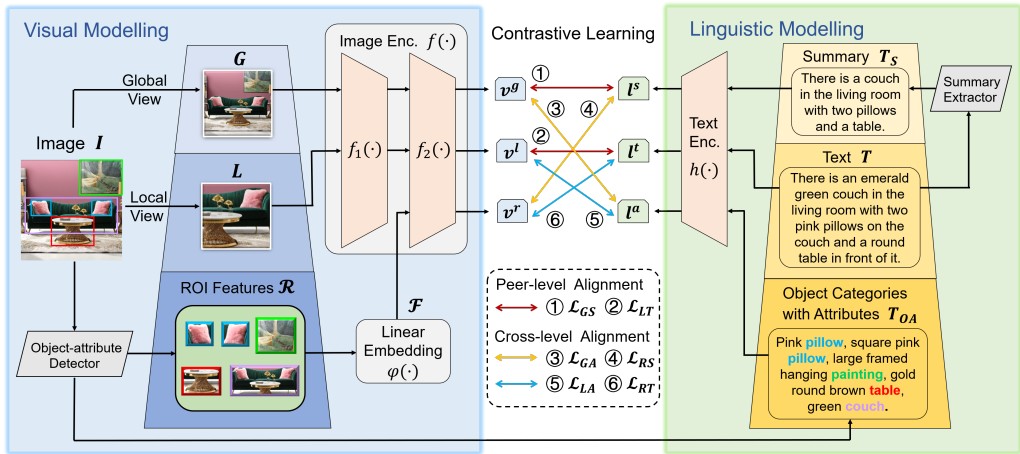

**Figure 2: Overall architecture of the proposed PyramidCLIP which is a dual-stream network.** The input elements of visual modelling and linguistic modelling both have three-level semantics. The elements of the two modalities interact through peer-level semantics alignment and cross-level relation alignment.

$l^t = h(T)$. Finally, $(\boldsymbol{v}^g, \boldsymbol{l}^s)$ and $(\boldsymbol{v}^l, \boldsymbol{l}^t)$ are pulled together through contrastive learning losses ① and ② respectively (refer to Figure 2), and other samples in the same batch are treated as negative samples. We term this contrasting process as *Peer-level Semantics Alignment*.

Furthermore, in order to explicitly model the relationship between salient objects in the image, the ROI feature sequence $\mathcal{R} = \{\boldsymbol{o}_1, \boldsymbol{o}_2, ..., \boldsymbol{o}_M\}$ of $M$ detected salient objects in the image $I$, with the category and attribute information for each object, are extracted through a pre-trained object-attribute detector. Then a linear embedding module $\varphi$ is used to transform $\mathcal{R}$ into the latent dimension of Multi-head Self-attention (MHSA) layer (12) in the image encoder. The feature sequence is successively fed into the rear part $f_2$, which contains one or more MHSA layers, to adaptively capture the relation between these salient instances, generating the final representation vector $\boldsymbol{v}^r$, *i.e.*, $\boldsymbol{v}^r = f_2(\varphi(\mathcal{R}))$. For the object categories with attributes, we join them together, constructing another text description $T_{\mathrm{OA}}$, to provide a more granular, comprehensive and accurate caption for the image. Then we feed it into the text encoder, generating the representation vector $\boldsymbol{l}^a = h(T_{\mathrm{OA}})$. To avoid the relation modelling being overwhelmed by the context semantic modelling, we have $(\boldsymbol{v}^g, \boldsymbol{l}^a)$, $(\boldsymbol{v}^r, \boldsymbol{l}^s)$, $(\boldsymbol{v}^l, \boldsymbol{l}^a)$ and $(\boldsymbol{v}^r, \boldsymbol{l}^t)$ as another four positive pairs, and the distances between which are narrowed through contrastive learning losses ③, ④, ⑤ and ⑥ respectively, termed as *Cross-level Relation Alignment*.

It is worth noting that at the inference stage, only the original image-text pair $(I, T)$ is used, *i.e.*, the visual representation $\boldsymbol{v}^i$ from $I$ and the linguistic representation $\boldsymbol{l}^t$ from $T$.

## 3.2 Peer-level Semantics Alignment

Now we present the details of the *peer-level semantics alignment*. As mentioned above, the dual-stream vision-language contrastive learning methods such as CLIP strongly rely on the assumption that the image-text pairs are of good quality of one-to-one correspondence. However, semantic mismatch between images and text captions often occurs in the automatically harvested data. Therefore, we construct an input pyramid with multi-level semantics on both sides of the dual-stream network, and then align image and text within the same semantics level. Specifically, the image $I$ is transformed to the global view $G$ and the local view $L$ through two random crops with different ratios. For the text caption, besides the original caption $T$, text summarization $T_{\mathrm{S}}$ with more compact semantics is extracted using a pre-trained text summarization extractor.

**Coarse-grained Global Contrast** We set the random crop ratio for generating global view $G$ to be $[0.9, 1]$, which basically contains all the information in the original image. Text summarization $T_{\mathrm{S}}$ condenses the original caption $T$, removing some redundant and overly detailed information in the $T$. $G$ and $T_{\mathrm{S}}$ both capture global information and can be used as pairs of positive samples. The projected embedding $\boldsymbol{v}^g$ and $\boldsymbol{l}^s$ of $G$ and $T_{\mathrm{S}}$ are pulled closer through contrastive learning.

**Fine-grained Local Contrast** Since the alignment of the global view $G$ with the text summarization $T_{\mathrm{S}}$ described above is relatively coarse, finer-grained information is largely discarded. Intuitively, some image sub-regions can be aligned with the original caption $T$. To this end, we introduce fine-grained local contrast. We set the random cropping ratio for generating local view $L$ to be $[0.5, 1]$, which focuses on the sub-region of the image $I$. Then the projected embeddings $\boldsymbol{v}^l$ and $\boldsymbol{l}^t$ of $L$ and $T$ are also brought together through contrastive loss (refer to Section 3.4).

Naturally, we have also tried to bring the finer-grained, peer-level $(\boldsymbol{v}^r, \boldsymbol{l}^a)$ closer, but there is no further gain (see Appendix F).

### 3.3  Cross-level Relation Alignment

To further improve the alignment precision, we introduce the ROI feature of each salient object in the image and the corresponding description with category and attributes, using a pre-trained object-attribute detector, as the most fine-grained semantic level to provide more accurate supervisions. Specifically, given an image $I$ with $M$ salient objects, the extracted visual semantics of each object region is formulated as $[\boldsymbol{o'_m}, \boldsymbol{z}_m]$, where $m$ denotes the $m_{th}$ object, $\boldsymbol{o'_m}$ is a 2048-dimensional feature vector and $\boldsymbol{z}_m$ is a 4-dimensional normalized position vector indicating the coordinates of top-left and bottom-right corners. By concatenating $\boldsymbol{o'_m}$ and $\boldsymbol{z}_m$, we have the 2052-dimensional position-sensitive ROI feature vector $\boldsymbol{o}_m$, forming the ROI feature sequence $\mathcal{R} = \{\boldsymbol{o}_1, \boldsymbol{o}_2, ..., \boldsymbol{o}_M\}$ with the order organized from high confidence to low. Then $\mathcal{R} \in \mathbb{R}^{M \times 2052}$ is transformed into $\mathbb{R}^{M \times d}$ using the projector in the embedding module $\varphi$, where $d$ indicates the latent dimension of the MHSA layers in the image encoder. A randomly initialized $d$-dimensional class token is additionally appended at the front, resulting $\mathcal{F} \in \mathbb{R}^{(1+M) \times d}$ , which is further feed into the rear part $f_2$ of the image encoder to compute the normalized ROI relation embedding $\boldsymbol{v}^r$, *i.e.*, $\mathcal{F} = \varphi(\mathcal{R})$ and $\boldsymbol{v}^r = f_2(\mathcal{F})$. Note that feature sequence $\mathcal{F}$ is position-sensitive following $\mathcal{R}$. Thus, positional embedding is not applied before it enters into the MHSA layer. Meanwhile, the detected names of category and attributes for each salient object form a phrase with one or more adjectives (attributes) modifying a noun (category), like "`gold round brown table`" in Figure 2. Then all the $M$ phrases from $M$ salient objects are joint into a text description $T_{\mathrm{OA}}$ with the same order as the ROI feature sequence, and the phrases are separated by commas. Next, $T_{\mathrm{OA}}$ is input to the text encoder to obtain the embedding $\boldsymbol{l}^a = h(T_{\mathrm{OA}})$.

To enhance the relation modelling capacity of the text encoder, while avoiding weakening the reasoning ability of the image encoder, $(\boldsymbol{v}^g, \boldsymbol{l}^a)$, $(\boldsymbol{v}^r, \boldsymbol{l}^s)$, $(\boldsymbol{v}^l, \boldsymbol{l}^a)$ and $(\boldsymbol{v}^r, \boldsymbol{l}^t)$ are used as another four positive pairs, and the distance between each pair is minimized simultaneously. Since the object-level inputs $\mathcal{R}$ and $T_{\mathrm{OA}}$ are direct concatenation of feature vectors and phrases respectively, hence very fine-grained, while other inputs $G$, $L$, $T_{\mathrm{S}}$ and $T$ are complete images or sentences, we term this contrasting process *cross-level relation alignment*.

In the case that the visual model is a convolution neural network (CNN), the traditional pooling layer is replaced by attention pooling, which actually is a MHSA layer. So the embedded ROI feature sequence $\mathcal{F}$ is input to the attention pooling layer, *i.e.*, $f_2$, which indicates the final attention pooling layer, as shown in Figure 3(a). For the transformer-based visual model (ViT), the sequence $\mathcal{F}$ can be directly input to a transformer layer. Considering that $\mathcal{F}$ already encodes high-level visual semantics, we feed it into the rear part $f_2$ of the ViT encoder, see Figure 3(b).

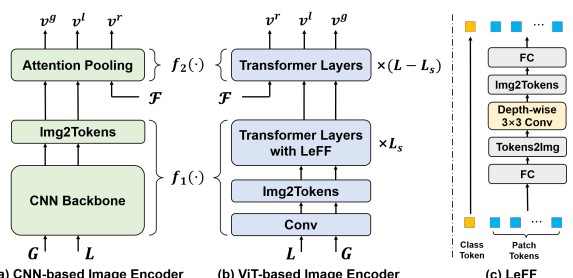

**Figure 3:** (a) The schematic of CNN-based image encoder. (b) The schematic of ViT-based image encoder. (c) The structure of LeFF module in ViT.

Besides, as pointed out in (20; 21), the standard ViT may not fully leverage the local context information, which limits the visual representation capacity of ViT-based image encoder. Following (21), we incorporate a depth-wise convolution into the Feed-Forward module of ViT, termed Locally-enhanced Feed-Forward (LeFF), improving the patch-level local perception and interaction. The structure of LeFF is shown in Figure 3(c). First, the patch tokens are projected into a higher dimension through a linear projection layer and reshaped. Next, a $3 \times 3$ depth-wise convolution is utilized to capture the local information. Then the feature maps are flattened to a token sequence and re-projected into

the initial dimension. While the class token is identical during the process and is concatenated with locally-enhanced patch tokens, generating the final output. As depicted in Figure 3(b), LeFF is only applied in the front part $f_1$ of the ViT-based image encoder, since it is clearly not suitable for the embedded ROI feature sequence $\mathcal{F}$. In Figure 3(b), $L$ denotes the total number of transformer layers in ViT and $L_s$ is the number of transformer layers with LeFF in $f_1$. And $L = 12$ and $L_s = 9$ in the experiments. The influence of different settings of $L_s$ can be seen in Appendix F.

### 3.4 Softened Objective Function

For a batch of $N$ image-text pairs $\{(I_i, T_i)\}_{i=1}^N$, where $i$ indicates the $i_{th}$ pair, the normalized embedded vectors $\{\boldsymbol{v}_i^g, \boldsymbol{v}_i^l, \boldsymbol{v}_i^r, \boldsymbol{l}_i^s, \boldsymbol{l}_i^t, \boldsymbol{l}_i^a\}_{i=1}^N$ of the same dimension are obtained by the dual-stream encoders. In this formulation, $\boldsymbol{v}_i^g$, $\boldsymbol{v}_i^l$ and $\boldsymbol{v}_i^r$ are generated by the image encoder from global-crop image $G$, local-crop image $L$ and ROI feature sequence $\mathcal{R}$ respectively, while $\boldsymbol{l}_i^s$, $\boldsymbol{l}_i^t$ and $\boldsymbol{l}_i^a$ are generated by the text encoder from text summarization $T_S$, original text $T$ and object-attribute description $T_{OA}$ respectively. Then we use this vector group to construct six supervision signals $\mathcal{L}_{GS}$, $\mathcal{L}_{LT}$, $\mathcal{L}_{GA}$, $\mathcal{L}_{RS}$, $\mathcal{L}_{LA}$ and $\mathcal{L}_{RT}$ for in-batch contrastive learning, which can be calculated with $\{(\boldsymbol{v}_i^g, \boldsymbol{l}_i^s)\}_{i=1}^N$, $\{(\boldsymbol{v}_i^l, \boldsymbol{l}_i^t)\}_{i=1}^N$, $\{(\boldsymbol{v}_i^g, \boldsymbol{l}_i^a)\}_{i=1}^N$, $\{(\boldsymbol{v}_i^r, \boldsymbol{l}_i^s)\}_{i=1}^N$, $\{(\boldsymbol{v}_i^l, \boldsymbol{l}_i^a)\}_{i=1}^N$ and $\{(\boldsymbol{v}_i^r, \boldsymbol{l}_i^t)\}_{i=1}^N$ respectively. Our six contrastive losses, with the formulation of InfoNCE (22), are designed to achieve the alignment between visual representation and linguistic representation from disparate semantic levels. Take the first loss term $\mathcal{L}_{GS}$ from $\{(\boldsymbol{v}_i^g, \boldsymbol{l}_i^s)\}_{i=1}^N$ as an example. For the $i_{th}$ pair, the normalized vision-to-language similarity $\boldsymbol{p}_i^v(G) = \{p_{ij}^v(G)\}_{j=1}^N$ and language-to-vision similarity $\boldsymbol{p}_i^l(T_S) = \{p_{ij}^l(T_S)\}_{j=1}^N$ can be calculated through:

$$p_{ij}^v(G) = \frac{\exp(\text{sim}(\boldsymbol{v}_i^g, \boldsymbol{l}_j^s)/\tau)}{\sum_{j=1}^N \exp(\text{sim}(\boldsymbol{v}_i^g, \boldsymbol{l}_j^s)/\tau)}, \quad p_{ij}^l(T_S) = \frac{\exp(\text{sim}(\boldsymbol{l}_i^s, \boldsymbol{v}_j^g)/\tau)}{\sum_{j=1}^N \exp(\text{sim}(\boldsymbol{l}_i^s, \boldsymbol{v}_j^g)/\tau)}, \quad (1)$$

where $\tau$ is a learnable temperature parameter initialized with $0.07$ and the function $\text{sim}(\cdot)$ conducts dot product to measure the similarity scores.

In standard practice, for the $i_{th}$ pair, the corresponding one-hot label vectors of the ground-truth $\boldsymbol{y}_i^v(G) = \{y_{ij}^v(G)\}_{j=1}^N$ and $\boldsymbol{y}_i^l(T_S) = \{y_{ij}^l(T_S)\}_{j=1}^N$, with positive pair denoted by $1$ and negatives by $0$, are used as the targets to calculate cross-entropy. This kind of hard targets assumes there is absolutely no similarity between unpaired image and text. However, within a large-size batch, unpaired image and text may have more or less local similarities, *i.e.*, some local regions in an image may be matched with some words or phrases in other unpaired texts. To address this problem for better generalization, we use label smoothing to soften the hard targets. The corresponding softened targets $\widetilde{\boldsymbol{y}}_i^v(G)$ and $\widetilde{\boldsymbol{y}}_i^l(T_S)$ for the $i_{th}$ pair can be formulated as:

$$\widetilde{\boldsymbol{y}}_i^v(G) = (1 - \alpha)\boldsymbol{y}_i^v(G) + \alpha/(N-1), \quad \widetilde{\boldsymbol{y}}_i^l(T_S) = (1 - \alpha)\boldsymbol{y}_i^l(T_S) + \alpha/(N-1), \quad (2)$$

where $\alpha$ is the smoothing hyper-parameter set to $0.2$ in our experiments. Then $\mathcal{L}_{GS}$ can be written as:

$$\mathcal{L}_{GS} = -\frac{1}{2N} \sum_{i=1}^N \sum_{j=1}^N (\widetilde{y}_{ij}^v(G) \cdot \log(p_{ij}^v(G)) + \widetilde{y}_{ij}^l(T_S) \cdot \log(p_{ij}^l(T_S))). \quad (3)$$

The other loss terms $\mathcal{L}_{LT}$, $\mathcal{L}_{GA}$, $\mathcal{L}_{RS}$, $\mathcal{L}_{LA}$ and $\mathcal{L}_{RT}$ can be calculated similarly. We then divide them into three groups that are respectively the peer-level alignment $\mathcal{L}_{peer} = (\mathcal{L}_{GS} + \mathcal{L}_{LT})/2$, the global-relation cross-level alignment $\mathcal{L}_{cross}^{global} = (\mathcal{L}_{GA} + \mathcal{L}_{RS})/2$ and the local-relation cross-level alignment $\mathcal{L}_{cross}^{local} = (\mathcal{L}_{LA} + \mathcal{L}_{RT})/2$. Therefore, the overall objective function of PyramidCLIP is:

$$\mathcal{L} = (1 - \lambda - \mu)\mathcal{L}_{peer} + \lambda\mathcal{L}_{cross}^{global} + \mu\mathcal{L}_{cross}^{local}, \quad (4)$$

where the loss weights $\lambda$ and $\mu$ are both set to $1/3$ in our experiments.

## 4 Experiments

### 4.1 Implementation Details and Datasets

**Pre-training Stage** We experiment on three different architectures, PyramidCLIP-ResNet50/ViT-B32/ViT-B16, according to the choice of image encoder. Their detailed architecture designs follow that of CLIP (6). LAION99M contains 99M image-text pairs with the highest similarity selected from LAION400M (23) according to the similarity scores provided by the producer. We use the publicly

released T5 model (19) to extract text summarization for all texts and utilize an object-attribute detector pre-trained by VinVL (16) to extract salient object features together with category and attribute information in the image. Please refer to Appendix A for details.

**Downstream Tasks** We validate the effectiveness of our proposed method on five downstream tasks: zero-shot image classification, zero-shot image-text retrieval, linear probe, object detection and instance segmentation. For classification, experiments are carried out on 11 datasets, such as ImageNet (13), CIFAR-100 (28). For image-text retrieval, experiments are conducted on Flickr30K (29) and MS-COCO (30). For object detection and instance segmentation, the proposed method is verified on PASCAL VOC (31) and MS-COCO. More details can be found in Appendix B.

**Table 1:** Pre-training datasets.

| Dataset | Size |
|---------|------|
| SBU (24) | 1M |
| CC3M (25) | 3M |
| CC12M (26) | 10M |
| YFCC15M-V1 (27) | 15M |
| YFCC15M-V2 (10) | 15M |
| LAION99M (23) | 99M |
| Total | 143M |

## 4.2 Fair Comparison against SoTA

We first compare our method against other SoTA approaches on ImageNet zero-shot classification task using the same amount of pre-training data YFCC15M-V2 and the results are shown in Table 2. It can be seen that compared to CLIP, PyramidCLIP improves the top-1 accuracy by 10.6%/13.2%/10.0% when the visual model is ResNet50/ViT-B32/ViT-B16 respectively. Furthermore, PyramidCLIP outperforms all other SoTA approaches by a large margin. In addition, since the distribution of different datasets can vary vastly, we also conduct experiments on YFCC15M-V1 and LAION15M, which is obtained by sampling 15 million image-text pairs from LAION99M for fair comparison. The results can be seen in Appendix C and our PyramidCLIP still shows great superiority.

**Table 2:** Zero-shot(ZS) classification results on ImageNet.

| Method | Image Encoder | ImageNet ZS Top1 |
|--------|---------------|------------------|
| CLIP (6) | ResNet50 | 37.2[†] |
| SLIP (32) | | 28.5[†] |
| FILIP (8) | | 21.3[†] |
| DECLIP (10) | | 44.4[†] |
| **PyramidCLIP** | | **47.8** |
| CLIP (6) | ViT-B/32 | 32.8[†] |
| SLIP(32) | | 34.3[†] |
| FILIP (8) | | 39.5[†] |
| DECLIP (10) | | 43.2[†] |
| DeFILIP (33) | | 45.0[†] |
| **PyramidCLIP** | | **46.0** |
| CLIP$^\diamond$ | ViT-B/16 | 40.7 |
| **PyramidCLIP** | | **50.7** |

$^\diamond$ Our Implementation  $^†$ Reported in (33)

## 4.3 Comparison on Large-scale Datasets

We further validate the effectiveness of our method on a large-scale dataset, *i.e.*, 143M image-text pairs, and downstream zero-shot image-text retrieval and image classification results are shown in Table 3. It can be seen that on image-text retrieval task, PyramidCLIP exceeds CLIP trained on 400M data and DECLIP by a large margin. And on ImageNet classification task, with the same amount of pre-training data, PyramidCLIP significantly exceeds the results of CLIP by 6.1%/3.8%/3.5% using ResNet50/ViT-B32/ViT-B16 as image encoder. Furthermore, it is worth noting that, when the vision model is ResNet50, PyramidCLIP trained on 143M data surpass CLIP using 400M data, which greatly improves data efficiency.

## 4.4 Transferability to Small-scale Classification Datasets

In this section, we validate the transferability of our method on 10 relatively small downstream classification datasets, both on zero-shot and linear probe tasks. The results are shown in Table 4. It can be seen that the average accuracy of PyramidCLIP on 10 datasets all exceed CLIP trained on 400M data on two kinds of tasks. It is worth noting that our pre-training data is less than 36% of CLIP, but the average accuracy is better, indicating higher data utilization.

## 4.5 Transferability to Object Detection and Instance Segmentation

In order to verify that our model can better exploit the relationship between objects in the image, we further validate our models on object detection and instance segmentation tasks. Specifically, we take the visual model ResNet50 to initialize the backbone of Faster R-CNN (34) and Mask R-CNN (35) and then all the parameters are fine-tuned. The results are shown in the Table 6. It can be seen that our model significantly outperforms random initialization and surpasses CLIP and DECLIP model.

**Table 3:** Zero-shot image-text retrieval results and image classification top-1 accuracy. IN denotes ImageNet.

| Method | Image Encoder | Pretrain Dataset | Flickr30K | | | | MS-COCO | | | | IN ZS Top1 |
|---|---|---|---|---|---|---|---|---|---|---|---|
| | | | I2T | | T2I | | I2T | | T2I | | |
| | | | R@1 | R@5 | R@1 | R@5 | R@1 | R@5 | R@1 | R@5 | |
| CLIP* | ResNet50 | 400M | 79.2 | 95.2 | 57.9 | 84.1 | 47.6 | 73.1 | 27.4 | 51.8 | 59.6 |
| DECLIP† | | 88M | 60.4 | 85.3 | 46.3 | 74.4 | 32.0 | 57.8 | 21.7 | 44.6 | **62.5** |
| CLIP◇ | | 143M | 80.6 | 95.7 | 63.6 | 87.3 | 51.8 | 76.4 | 34.0 | 60.0 | 55.3 |
| **PyramidCLIP** | | 143M | **86.3** | **98.0** | **71.6** | **91.3** | **55.0** | **79.8** | **39.6** | **66.2** | 61.4 |
| CLIP* | ViT-B/32 | 400M | 77.6 | 93.6 | 59.0 | 83.7 | 49.2 | 74.1 | 29.8 | 54.4 | 63.2 |
| DECLIP† | | 88M | 59.8 | 84.4 | 46.2 | 74.5 | 32.6 | 59.1 | 22.1 | 45.8 | **66.6** |
| CLIP◇ | | 143M | 81.3 | 95.4 | 63.3 | 87.0 | 51.1 | 76.4 | 34.4 | 60.6 | 58.0 |
| **PyramidCLIP** | | 143M | **84.2** | **96.4** | **69.1** | **89.8** | **52.8** | **78.1** | **38.8** | **64.9** | 61.8 |
| CLIP* | ViT-B/16 | 400M | 84.6 | 97.3 | 65.0 | 87.8 | 51.7 | 76.1 | 32.5 | 57.7 | **68.8** |
| CLIP◇ | | 143M | 84.5 | 97.4 | 70.5 | 90.9 | **56.9** | 79.6 | 38.8 | 65.0 | 63.4 |
| **PyramidCLIP** | | 143M | **85.6** | **97.7** | **74.5** | **92.9** | 55.7 | **80.8** | **42.6** | **68.6** | 66.9 |

\* Tested with the released model: https://github.com/openai/CLIP#api  ◇ Our Implementation
† Tested with: https://github.com/Sense-GVT/DeCLIP#our-pretrain-declip-model-w-text-encoder

**Table 4:** Accuracy on 10 datasets with ResNet50 image encoder. C10/100/F101/FLOW/CAL/AIR is CIFAR-10/CIFAR-100/Food101/Flowers/Caltech/Aircraft. AVG represents average accuracy across 10 datasets.

| Task | Method | Pretrain Dataset | PETS | C10 | C100 | DTD | CARS | F101 | FLOW | AIR | SUN | CAL | AVG |
|---|---|---|---|---|---|---|---|---|---|---|---|---|---|
| Zero Shot | CLIP | 400M | **85.4** | 75.6 | 41.6 | 41.7 | 55.8 | **81.1** | **65.9** | **19.3** | 59.6 | **82.1** | 60.8 |
| | CLIP◇ | 143M | 77.0 | 56.4 | 26.7 | 41.4 | 54.6 | 69.8 | 60.4 | 7.3 | 60.6 | 76.1 | 53.6 |
| | **PyramidCLIP** | 143M | 83.7 | **81.5** | **53.7** | **47.2** | **65.0** | 67.8 | 65.8 | 12.6 | **65.8** | 81.7 | **62.4** |
| Linear Probe | CLIP | 400M | 85.1 | 88.7 | 70.3 | 76.4 | 78.3 | **86.4** | 96.1 | 49.1 | 73.3 | 89.6 | 79.3 |
| | DECLIP | 88M | **88.7** | 89.8 | 71.2 | **76.8** | 81.7 | 82.7 | **99.2** | 48.4 | 72.8 | 93.9 | 80.5 |
| | CLIP◇ | 143M | 82.9 | 84.5 | 64.5 | 74.3 | 79.3 | 80.5 | 93.2 | 45.4 | 74.7 | 91.5 | 77.1 |
| | **PyramidCLIP** | 143M | 87.8 | **91.8** | **75.6** | 75.8 | **81.8** | 81.9 | 93.0 | **53.1** | 76.1 | **94.2** | **81.3** |

◇ Our Implementation

## 4.6 Ablation Study

In this section, we verify the effectiveness of each component in PyramidCLIP on downstream zero-shot ImageNet classification task, and all the experiments are pre-trained for 8 epochs on YFCC15M-V1. The results are listed in Table 5, which indicate that on the basis of the peer-level alignment, all the other components including cross-level global-relation and local-relation alignment, LeFF in ViT and softened objectives can bring significant gains individually. More ablation results can be seen in Appendix F, including the detailed ablation on $L_s$, the choice of peer-level alignment and the effectiveness each component on downstream object detection task.

**Table 5:** Ablation study of each component on ImageNet zero-shot classification task. "Soften" means all the objectives are softened.

| Image Encoder | Components | | | | | ImageNet ZS Top1 |
|---|---|---|---|---|---|---|
| | $\mathcal{L}_{peer}$ | $\mathcal{L}_{cross}^{global}$ | $\mathcal{L}_{cross}^{local}$ | Soften | LeFF | |
| ResNet50 | ✓ | | | | - | 32.8 |
| | ✓ | ✓ | | | - | 35.0(+2.2) |
| | ✓ | ✓ | ✓ | | - | 36.7(+3.9) |
| | ✓ | ✓ | ✓ | ✓ | - | **38.6(+5.8)** |
| ViT-B/32 | ✓ | | | | | 28.8 |
| | ✓ | ✓ | | | | 32.1(+3.3) |
| | ✓ | ✓ | ✓ | | | 33.4(+4.6) |
| | ✓ | ✓ | ✓ | ✓ | | 35.0(+6.2) |
| | ✓ | ✓ | ✓ | ✓ | ✓ | **35.9(+7.1)** |

## 4.7 Visualization

**Semantic Features** We utilize $t$-SNE (36) to visualize the learned semantic features of CIFAR-10 (28). Each text feature of 10 categories is obtained using the ensemble of 80 prompt templates. And image features are extracted with ResNet50 visual encoder. As depicted in Figure 4, CLIP pre-trained on 143M data has a poor clustering performance, with the visual features of most categories overlapping heavily and the textual features of some two categories being very close, such as (`automobile`, `truck`) and (`frog`, `deer`). Although CLIP pre-trained on 400M data (6) performs better than CLIP on 143M data, the visual features of some categories like `dog` and `cat` still have a large overlap. In comparison, the semantic features extracted by PyramidCLIP on 143M data, are well separated with each textual feature surrounded by most visual features of the same category.

**Table 6:** Object detection and instance segmentation results on VOC and COCO with ResNet50 as backbone.

| Initialized Weights | Pre-train Dataset | Object Detection | | | | | | Instance Segmentation | | |
| --- | --- | --- | --- | --- | --- | --- | --- | --- | --- | --- |
| | | VOC | | | COCO | | | COCO | | |
| | | $AP^{bb}$ | $AP_{50}^{bb}$ | $AP_{75}^{bb}$ | $AP^{bb}$ | $AP_{50}^{bb}$ | $AP_{75}^{bb}$ | $AP^{mk}$ | $AP_{50}^{mk}$ | $AP_{75}^{mk}$ |
| Random | - | 26.5 | 51.6 | 22.8 | 28.5 | 46.2 | 29.8 | 25.6 | 43.4 | 26.8 |
| CLIP (6) | 400M | 45.5 | 73.5 | 47.9 | 36.5 | 56.1 | 38.8 | 31.9 | 52.7 | 33.5 |
| DECLIP (10) | 88M | 50.0 | 77.4 | 53.6 | 37.4 | 57.2 | 40.1 | 32.5 | 53.6 | 34.4 |
| **PyramidCLIP** | 143M | **50.9** | **78.7** | **54.7** | **38.0** | **58.1** | **40.8** | **33.0** | **54.6** | **35.1** |

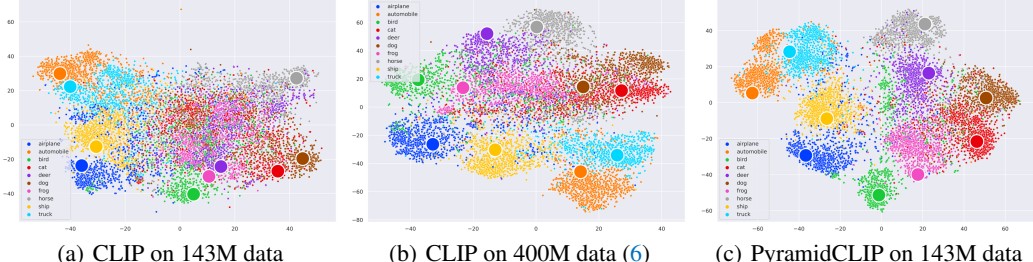

| (a) CLIP on 143M data | (b) CLIP on 400M data (6) | (c) PyramidCLIP on 143M data |

**Figure 4:** Visualization of semantic features of CIFAR-10 test set. Big points represent text features and small points indicate image features. Different colors represent different categories.

**Grad-CAM Heatmaps** Grad-CAM (37) is also utilized to help understand why PyramidCLIP outperforms CLIP. Specifically, we conduct this through text-to-image retrieval on MS-COCO, using CLIP/PyramidCLIP-ResNet50 pre-trained on 143M data. For each query text, the model retrieves top5 images with highest similarities. Then for each of the five retrieved images, we use Grad-CAM to find which areas have the highest activation to the query text. As shown in Figure 5, PyramidCLIP has better retrieval performance than CLIP. Moreover, PyramidCLIP can accurately capture the complete object regions that are highly matched with the query texts, while CLIP captures regions with either components missing or additional noises, or even obtains completely irrelevant matches. More visualization results can be found in Appendix G.

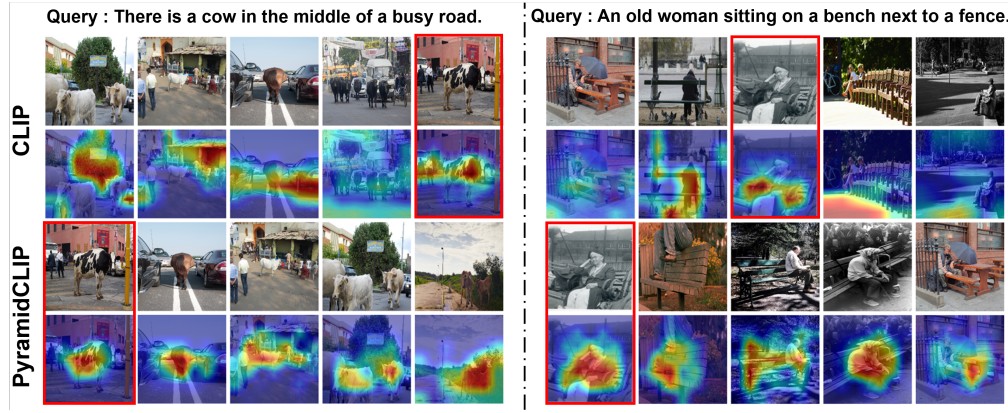

**Figure 5:** Grad-CAM heatmaps for top5 retrieved images. From left to right are images from rank1 to rank5. Red box indicates the groundtruth image matched with the query text.

## 5 Conclusion

In this paper, we have proposed a hierarchical pre-training method, termed PyramidCLIP, to achieve improved alignment between visual and linguistic modalities. It resolves the issue that the webly-crawled data is not in perfect one-to-one correspondence by explicitly constructing pyramidal semantic inputs at the both sides of dual-stream network. We also show that softened peer-level semantics alignment and cross-level relation alignment can interact between two modalities and are beneficial. PyramidCLIP achieves the state-of-the-art results on five downstream tasks, which demonstrates the superiority.

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
