# PyramidCLIP: Hierarchical Feature Alignment for Vision-language Model Pretraining

## A  Pre-training Stage Settings

### A.1  Datasets

In Section 4.1, we list all the subsets of the total 143M dataset, and here we introduce them in detail. SBU [1] is a relatively small image-text dataset, which contains 1 million image-text pairs obtained from Flickr. YFCC15M [2] is a commonly used subset of YFCC100M [2] and there are mainly two versions of YFCC15M, V1 and V2. YFCC15M-V1 is obtained by applying the same filtering rule on YFCC100M as CLIP [3], while YFCC15M-V2 is collected by DeCLIP [4] with a different filtering strategy. In addition to a subset of YFCC, the V2 dataset also contains some additional data crawled from the Internet and is of higher quality than V1. CC3M [5] and CC12M [6] conduct the same image-text filter pipeline on Internet webpage sources, and the difference is that the filtering method of the latter is more relaxed. LAION400M [7] is one of the largest openly available image-text datasets. We rank the image-text pairs by their similarity scores, which are pre-computed by the producer using a pre-trained CLIP model, and pick up a 99M subset with the highest similarity scores. Combining all the above datasets, we finally have a dataset of 143M image-text pairs. During the training process, we randomly shuffle the training sequence of datasets at the beginning of each training epoch, then train our model on these datasets one by one.

### A.2  Implementation Details

**Model Architectures** We follow the same architecture design as CLIP [3] for PyramidCLIP-ResNet50/ViT-B32/ViT-B16, except that we incorporate a depth-wise convolution into the Feed-Forward module of ViT, namely, LeFF. The input resolution of image encoder is $224 \times 224$ and the maximum context length of text encoder is 77. The final image and text features are projected to the same dimension, which is 1024 for PyramidCLIP-ResNet50 and 512 for PyramidCLIP-ViT, followed by L2 normalization before interaction.

**Details of the Object-attribute Detector** The object-attribute detector, adopting the framework of Faster R-CNN [8], is pre-trained by VinVL [9]. And image is resized to the resolution of $640 \times 640$ before entering the detector. We take 10 objects with the highest confidence from the detector to obtain the corresponding ROI features and category descriptions with attribute information.

**Pre-training Setup** We train our PyramidCLIP using an AdamW [10] optimizer and the cosine learning rate scheduler with a linear warmup. Specifically, the learning rate linearly increases from 0 to the peak value within $10\%$ of the total steps, and then decreases with a cosine anneal strategy. The weight decay rate of AdamW is set to 0.2. To save GPU memory, automatic mixed-precision [11] is used. The models are trained from scratch for either 8 or 32 epochs in our experiments, *i.e.*, 8 epochs for ablation and 32 epochs for comparison. When training on 15M datasets, including YFCC15M-V1, V2 and LAION15M, the batch size is set to 4096 and the peak learning rate is set to $2 \times 10^{-3}$. While on the 143M large-scale dataset, the batch size is set to 8192 and the peak learning rate is set to

Submitted to 36th Conference on Neural Information Processing Systems (NeurIPS 2022). Do not distribute.

$5 \times 10^{-4}$. Besides, PyramidCLIP-ResNet50/ViT-B32 takes 64 V100 GPUs to train on 143M data, while PyramidCLIP-ViT-B16 takes 128 A100 GPUs.

# B  Downstream Settings

## B.1  Datasets for Downstream Classification Task

Besides ImageNet [12], we also evaluate the transferability of our model on other 10 downstream classification datasets including Oxford-IIIT Pets [13], CIFAR-10 [14], CIFAR-100 [14], Describable Textures [15], Stanford Cars[16], Food-101 [17], Oxford Flowers 102 [18], FGVC Aircraft [19], SUN397 [20] and Caltech-101 [21]. The details of each dataset are listed in Table 1 and we follow the same data split and evaluation metric as CLIP for fair comparison.

Table 1: Datasets for downstream classification task.

| Dataset | Abbreviation | Classes | Train Size | Test Size | Evaluation Metric |
|---------|--------------|---------|------------|-----------|-------------------|
| CIFAR-10 | C10 | 10 | 50,000 | 10,000 | accuracy |
| CIFAR-100 | C100 | 100 | 50,000 | 10,000 | accuracy |
| Describable Textures | DTD | 47 | 3,760 | 1,880 | accuracy |
| Stanford Cars | CARS | 196 | 8,144 | 8,041 | accuracy |
| Food-101 | F101 | 101 | 75,750 | 25,250 | accuracy |
| Oxford-IIIT Pets | PETS | 37 | 3,680 | 3,669 | mean per class |
| Oxford Flowers 102 | FLOW | 102 | 2,040 | 6,149 | mean per class |
| FGVC Aircraft | AIR | 100 | 6,667 | 3,333 | mean per class |
| SUN397 | SUN | 397 | 19,850 | 19,850 | accuracy |
| Caltech-101 | CAL | 102 | 3,060 | 6,085 | mean per class |
| ImageNet | IN | 1000 | 1,281,167 | 50,000 | accuracy |

## B.2  Implementation Details

**Downstream Zero-shot Image Classification** Due to the fact that the labels of common classification datasets are mostly nouns rather than natural language descriptions, we adopt the same prompt setting as used in CLIP, that is, for every single class name, we generate 80 different textual descriptions with 80 prompt templates (such as "a photo of label"). The ensembles of these textual representations are used in computing similarities between images and label names.

**Downstream Zero-shot Image-text Retrieval** The image-text retrieval task can be split into two sub-tasks, *i.e.*, image retrieval and text retrieval, according to the target modality. We evaluate the zero-shot image-text retrieval capabilities on Flickr30K and MS-COCO dataset, which is performed by ranking image-text pairs according to their similarity scores.

**Downstream Image Linear Probe** To implement linear probe evaluation, we follow CLIP [3] to train a logistic regression classifier on the frozen visual features extracted by the image encoder. Specifically, we train the logistic regression classifier using L-BFGS algorithm provided by scikit-learn with maximum 1,000 iterations, and report the corresponding metric for each dataset. Moreover, the L2 regularization strength $C$ is determined using hyperparameter sweep on the validation sets.

**Downstream Object Detection and Instance Segmentation** Following [22, 23], for the downstream object detection and instance segmentation tasks, all the parameters are fine-tuned. For detection task on PASCAL VOC, the detector is trained for 24k steps with a batch size of 40, and the initial learning rate is 0.02 with 100 warm-up iterations and decays by 10 at 18k, 22k steps. The scale of image is randomly sampled from $[480, 800]$ with interval 32 during training and is set to 800 for inference. For the detection and instance segmentation on COCO, the model is trained for 90k iterations with the initial learning rate 0.02, and the scales of images are randomly sampled from $[600, 800]$ during training and is also set to 800 for inference.

**Downstream Image End-to-end Fine-tuning** In addition to the five downstream tasks mentioned in the main text, following MAE [24], we also conduct the end-to-end fine-tuning experiments on downstream classification task. Here, we elaborate the implementation details of this setting, and the corresponding results can be seen in Appendix E. For downstream end-to-end fine-tuning, we first

fine-tune the classifier layer alone while freezing the others for 8 epochs to endow the model a proper
initialization. Then we fine-tune all the parameters in the usual manner. AdamW is used as optimizer,
and the learning rate is set to 1e-3 and adjusted by a cosine learning rate scheduler without warmup.
We totally fine-tune the model for 128 epochs, including the initialization stage.

# C   Fair Comparison on YFCC15M-V1 and LAION15M

In this section, we compare our PyramidCLIP against CLIP under the same amount of pre-training
data, *i.e.*, YFCC15M-V1 and LAION15M. The results on downstream zero-shot image-text retrieval
task and ImageNet classification are shown in Table 2 and Table 3. It can be seen that PyramidCLIP
surpasses CLIP regardless of the distribution of pre-training dataset, demonstrating the superiority
of our pre-training method. Furthermore, we validate the effectiveness of proposed method on
downstream object detection and instance segmentation tasks, and the results are shown in Table 4. It
can be seen that the weights of our model exceeds that of CLIP model on both object detection task
and instance segmentation task. Noting, on object detection task, the improvement on PASCAL VOC
is more obvious than that on MS-COCO, since the amount of PASCAL VOC is smaller than COCO
and the effect of weight initialization is more conspicuous.

**Table 2:** Zero-shot image-text retrieval results on MS-COCO and zero-shot top1 accuracy on ImageNet. All the
models are pre-trained on YFCC15M-V1 for 32 epochs, except SLIP [25] for 100 epochs.

| Method | Image Encoder | MS-COCO | | | | ImageNet |
|---|---|---|---|---|---|---|
| | | I2T R@1 | I2T R@5 | T2I R@1 | T2I R@5 | ZS Top1 |
| CLIP$^\diamond$ | ResNet50 | 29.8 | 56.9 | 19.3 | 40.9 | 36.8 |
| **PyramidCLIP** | ResNet50 | **39.9** | **66.2** | **24.9** | **49.3** | **43.7** |
| CLIP$^\diamond$ | ViT-B/32 | 24.2 | 48.3 | 14.0 | 33.1 | 31.2 |
| **PyramidCLIP** | ViT-B/32 | **34.2** | **60.2** | **21.1** | **44.0** | **41.7** |
| CLIP$^\diamond$ | ViT-B/16 | 30.3 | 56.1 | 18.9 | 40.0 | 36.9 |
| SLIP$^\dagger$ [25] | ViT-B/16 | 33.9 | 60.0 | 22.5 | 45.4 | 45.0 |
| **PyramidCLIP** | ViT-B/16 | **38.2** | **65.0** | **25.0** | **49.3** | **46.0** |

$\diamond$ Our Implementation
† Tested with released model: https://github.com/facebookresearch/SLIP#vit-base

**Table 3:** Zero-shot image-text retrieval results on MS-COCO and zero-shot top1 accuracy on ImageNet. All the
models are pre-trained on LAION15M for 32 epochs.

| Method | Image Encoder | MS-COCO | | | | ImageNet |
|---|---|---|---|---|---|---|
| | | I2T R@1 | I2T R@5 | T2I R@1 | T2I R@5 | ZS Top1 |
| CLIP$^\diamond$ | ResNet50 | 31.5 | 57.0 | 18.9 | 39.8 | 35.6 |
| **PyramidCLIP** | ResNet50 | **33.3** | **60.4** | **24.4** | **48.4** | **41.9** |
| CLIP$^\diamond$ | ViT-B/32 | 25.9 | 49.8 | 15.6 | 34.8 | 32.6 |
| **PyramidCLIP** | ViT-B/32 | **28.5** | **55.5** | **21.2** | **43.7** | **39.9** |
| CLIP$^\diamond$ | ViT-B/16 | 31.4 | 56.2 | 19.2 | 40.6 | 38.3 |
| **PyramidCLIP** | ViT-B/16 | **32.2** | **60.7** | **25.6** | **49.8** | **45.3** |

$\diamond$ Our Implementation

# D   Downstream Task: Linear Probe

We first validate the transferability of our method on downstream classification task via linear probe.
The results are exhibited in Table 5. It can be seen that when the image encoder is ViT-B/32, the
average accuracy of PyramidCLIP pre-trained on 143M data exceeds that of CLIP using 400M data.
Furthermore, regardless of the image encoder used, our method outperforms CLIP on more than half
of the small datasets, noting that the amount of pre-training data we used is only about 36% of that
used by CLIP, further demonstrating the effectiveness of the proposed method.

**Table 4:** Fair comparison on object detection and instance segmentation tasks with ResNet50 as backbone.

| Initialized Weights | Pre-train Dataset | Object Detection | | | | | | Instance Segmentation | | |
|---|---|---|---|---|---|---|---|---|---|---|
| | | VOC | | | COCO | | | COCO | | |
| | | $AP^{bb}$ | $AP^{bb}_{50}$ | $AP^{bb}_{75}$ | $AP^{bb}$ | $AP^{bb}_{50}$ | $AP^{bb}_{75}$ | $AP^{mk}$ | $AP^{mk}_{50}$ | $AP^{mk}_{75}$ |
| CLIP$^{\diamond}$ | YFCC15M-V1 | 46.0 | 74.0 | 48.2 | 35.4 | 54.8 | 37.9 | 30.9 | 51.5 | 32.7 |
| **PyramidCLIP** | YFCC15M-V1 | **49.8** | **77.7** | **53.5** | **37.1** | **57.1** | **39.9** | **32.3** | **53.4** | **34.1** |
| CLIP$^{\diamond}$ | LAION15M | 46.8 | 74.9 | 49.5 | 35.5 | 54.9 | 37.9 | 30.9 | 51.4 | 32.4 |
| **PyramidCLIP** | LAION15M | **49.7** | **77.7** | **53.3** | **36.5** | **56.1** | **38.9** | **31.9** | **52.7** | **33.5** |

$^{\diamond}$ Our Implementation

**Table 5:** Linear probe accuracy on 10 datasets. C10/100/F101/FLOW/CAL/AIR is CIFAR-10/CIFAR-100/Food101/Flowers/Caltech/Aircraft. AVG represents average accuracy across 10 datasets.

| Method | Image Encoder | Pretrain Dataset | PETS | C10 | C100 | DTD | CARS | F101 | FLOW | AIR | SUN | CAL | AVG |
|---|---|---|---|---|---|---|---|---|---|---|---|---|---|
| CLIP$^{\star}$ | ViT-B/32 | 400M | 85.3 | 95.1 | 80.5 | 76.5 | 81.8 | **88.8** | **96.9** | **52.0** | 76.6 | 93.0 | 82.7 |
| **PyramidCLIP** | ViT-B/32 | 143M | **87.8** | **96.0** | **82.5** | **77.3** | **82.6** | 83.3 | 93.9 | 50.2 | **77.5** | **96.4** | **82.8** |
| CLIP$^{\star}$ | ViT-B/16 | 400M | **93.1** | 96.2 | 83.1 | 79.2 | 86.7 | **92.8** | **98.1** | **59.5** | 78.4 | 94.7 | **86.2** |
| **PyramidCLIP** | ViT-B/16 | 143M | 90.3 | **96.5** | **83.5** | **79.3** | **86.9** | 88.1 | 95.6 | 56.5 | **79.9** | **96.5** | 85.3 |

$^{\star}$ Tested with the released model: https://github.com/openai/CLIP#api

# E  Downstream Task: End-to-end Fine-tuning

On the basis of linear probe, we further validate the transferability of our method via end-to-end fine-tuning. The results are shown in Table 6. We compare PyramidCLIP against CLIP and supervised counterpart. It can be seen that PyramidCLIP pre-trained on 143M data exceeds both CLIP and supervised ResNet50. Also, it is worth noting that compared to CLIP, we use only 36% pre-training data, and compared with ResNet50 trained on manually-labeled ImageNet-1K, we didn't use any manually-labeled data.

**Table 6:** End-to-end fine-tuning accuracy on 11 downstream classification datasets with ResNet50 backbone. C10/C100/F101/FLOW/CAL/AIR/IN is CIFAR-10/CIFAR-100/Food101/Flowers/Caltech/Aircraft/ImageNet1k. AVG represents average accuracy across 11 datasets. Supervised(IN1K) denotes the model is supervised trained on ImageNet-1K dataset.

| Method | Pretrain Dataset | PETS | C10 | C100 | DTD | CARS | F101 | FLOW | AIR | SUN | CAL | IN | AVG |
|---|---|---|---|---|---|---|---|---|---|---|---|---|---|
| Supervised(IN1K)$^{\dagger}$ | 1.2M | **93.0** | 94.0 | **77.8** | 68.4 | 65.6 | 81.3 | **89.7** | **60.0** | 62.3 | 90.8 | 76.2 | 78.1 |
| CLIP$^{\star}$ | 400M | 74.5 | 95.0 | 69.4 | 70.4 | 73.4 | 86.4 | 88.4 | 57.8 | 65.4 | 89.7 | 76.6 | 77.0 |
| **PyramidCLIP** | 143M | 75.4 | **95.2** | 74.8 | **72.0** | **77.7** | **86.7** | 87.7 | 58.4 | **68.6** | **92.0** | **78.0** | **78.8** |

$^{\star}$ Initialized with the released model: https://github.com/openai/CLIP#api

$^{\dagger}$ Initialized with model from torchvision: https://download.pytorch.org/models/resnet50-0676ba61.pth

# F  More Ablation

In this section, we supplement the ablation study of some important components in PyramidCLIP. All the ablation experiments are conducted on YFCC15M-V1 and trained for 8 epochs.

## F.1  Supplementary Ablation of PyramidCLIP Components

**Ablation of Each Component on Other Downstream Tasks** In Section 4.6, we only provide ablation study of each component on ImageNet zero-shot classification. Here we list the corresponding ablation results on MS-COCO zero-shot image-text retrieval and PASCAL VOC object detection in Table 7, which indicate that on the basis of the peer-level alignment, all the other components in PyramidCLIP can still bring accuracy improvement individually on the two downstream tasks.

**Table 7:** Ablation study of each component on MS-COCO zero-shot image-text retrieval and PASCAL VOC object detection. "Soften" means all the objectives are softened.

| Image Encoder | Components | | | | | MS-COCO | | PASCAL VOC | |
|---|---|---|---|---|---|---|---|---|---|
| | $\mathcal{L}_{\text{peer}}$ | $\mathcal{L}_{\text{cross}}^{\text{global}}$ | $\mathcal{L}_{\text{cross}}^{\text{local}}$ | Soften | LeFF | I2T R@1 | T2I R@1 | $AP^{bb}$ | $AP^{bb}_{50}$ |
| ResNet50 | ✓ | | | | - | 28.5 | 16.6 | 45.7 | 74.3 |
| | ✓ | ✓ | | | - | 31.9(+3.4) | 18.5(+1.9) | 46.5(+0.8) | 75.1(+0.8) |
| | ✓ | ✓ | ✓ | | - | 34.6(+6.1) | 19.6(+3.0) | 47.0(+1.3) | 75.6(+1.3) |
| | ✓ | ✓ | ✓ | ✓ | - | **36.4(+7.9)** | **21.1(+4.5)** | **47.4(+1.7)** | **76.0(+1.7)** |
| ViT-B/32 | ✓ | | | | | 24.3 | 13.4 | - | - |
| | ✓ | ✓ | | | | 27.9(+3.6) | 16.0(+2.6) | - | - |
| | ✓ | ✓ | ✓ | | | 29.3(+5.0) | 17.7(+4.3) | - | - |
| | ✓ | ✓ | ✓ | ✓ | | 31.4(+7.1) | 18.2(+4.8) | - | - |
| | ✓ | ✓ | ✓ | ✓ | ✓ | **31.7(+7.4)** | **18.8(+5.4)** | - | - |

**The Influence of Different $L_s$ Settings** We further probe into the influence of the transformer layers $L_s$ in the front part of ViT-based image encoder on zero-shot ImageNet classification task. Note that the total number of transformer layers $L$ in ViT is 12. The corresponding results with different $L_s$ values are shown in Figure 1. It can be found that $L_s = 9$ achieves the best result, hence $L_s$ is set to 9 in our experiments. Besides, $L_s = 0$ represents that the feature sequence $\mathcal{F}$ is feed into the first transformer layer of ViT and all the 12 layers are without LeFF. While $L_s = 12$ indicates that all the 12 layers are with LeFF and the raw sequence $\mathcal{F}$ is directly input to the final projector without being processed by transformer, which is the reason why $L_s = 12$ shows such poor performance.

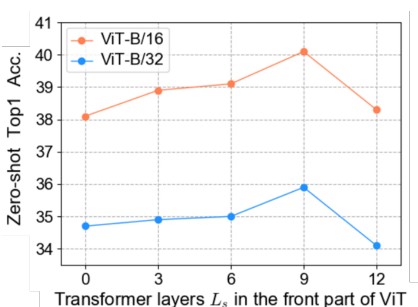

**Figure 1:** Zero-shot performance with different settings of $L_s$.

## F.2  Ablation of Other Possible Alignments

In addition to the current six loss terms of PyramidCLIP described in the main text, *i.e.*, the peer-level semantics alignment $\mathcal{L}_{\text{GS}}$ and $\mathcal{L}_{\text{LT}}$, the global-relation cross-level alignment $\mathcal{L}_{\text{GA}}$ and $\mathcal{L}_{\text{RS}}$, and the local-relation cross-level alignment $\mathcal{L}_{\text{LA}}$ and $\mathcal{L}_{\text{RT}}$, there are three other possible losses that are $\mathcal{L}_{\text{RA}}$, $\mathcal{L}_{\text{GT}}$ and $\mathcal{L}_{\text{LS}}$, corresponding to $(\boldsymbol{v}^r, \boldsymbol{l}^a)$, $(\boldsymbol{v}^g, \boldsymbol{l}^t)$ and $(\boldsymbol{v}^l, \boldsymbol{l}^s)$ respectively, as depicted in Figure 2(a). Note that $\mathcal{L}_{\text{RA}}$ belongs to the peer-level alignment, and $\mathcal{L}_{\text{GT}}$ and $\mathcal{L}_{\text{LS}}$ are semantically mismatched. Among them, $\mathcal{L}_{\text{GT}}$ is actually the original CLIP loss, shown in Figure 2(b). We will discuss the influence of these three losses in next two parts and explain why they are not incorporated into our PyramidCLIP paradigm.

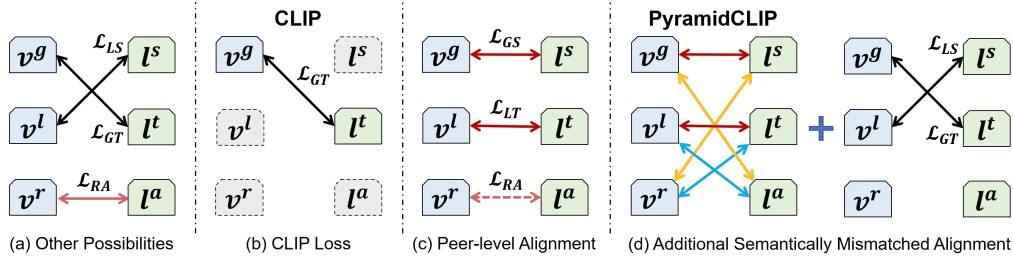

(a) Other Possibilities  (b) CLIP Loss  (c) Peer-level Alignment  (d) Additional Semantically Mismatched Alignment

**Figure 2:** Schematic diagram of various losses. (a) Other three possible losses besides PyramidCLIP. (b) The original CLIP loss. (c) Peer-level alignment. $\mathcal{L}_{\text{RA}}$ also belongs to it but is not included in PyramidCLIP. (d) Adding semantically mismatched alignment $\mathcal{L}_{\text{GT}}$ and $\mathcal{L}_{\text{LS}}$ into PyramidCLIP.

**Granular Ablation of Peer-level Alignment** In this part, we explore the influence of each sub-component belonging to the peer-level alignment, including $\mathcal{L}_{\text{GS}}$, $\mathcal{L}_{\text{LT}}$ and $\mathcal{L}_{\text{RA}}$, shown in Figure 2(c), and compared with original CLIP loss. As listed in Table 8, reducing the image random crop ratio of

CLIP can improve the model performance, *i.e.*, using $\mathcal{L}_{\mathrm{LT}}$ rather than $\mathcal{L}_{\mathrm{GT}}$ (see yellow rows), since the local view statistically removes some redundant information in the image, *i.e.*, some sub-regions not described in the text. That is why we use the image local view and the original text as a pair to construct the peer-level local contrast. Besides, it can be seen that the addition of global contrast $\mathcal{L}_{\mathrm{GS}}$ also brings significant improvement. However, there is no further gain when bringing into $\mathcal{L}_{\mathrm{RA}}$ (see blue rows), hence it is not included in PyramidCLIP. And we attribute this to the inherent precise alignment between the ROI feature sequence $\mathcal{R}$ and the object-attribute description $T_{\mathrm{OA}}$, since the feature and category with attributes of each salient object in the image are extracted in pairs by the pre-trained powerful object-attribute detector.

**Table 8:** Granular ablation results of peer-level alignment compared to the original CLIP loss.

| Image Encoder | Method | ImageNet | MS-COCO | | PASCAL VOC | |
|---|---|---|---|---|---|---|
| | | ZS Top1 | I2T R@1 | T2I R@1 | $AP^{bb}$ | $AP^{bb}_{50}$ |
| ResNet50 | $\mathcal{L}_{\mathrm{GT}}$ (CLIP) | 30.0 | 26.0 | 13.5 | 44.6 | 73.4 |
| | $\mathcal{L}_{\mathrm{LT}}$ | 30.8(+0.8) | 26.5(+0.5) | 14.0(+0.5) | 44.9(+0.3) | 73.5(+0.1) |
| | $\mathcal{L}_{\mathrm{LT}} + \mathcal{L}_{\mathrm{GS}}$ | 32.8(+2.8) | 28.5(+2.5) | 16.6(+3.1) | 45.7(+1.1) | 74.3(+0.9) |
| | $\mathcal{L}_{\mathrm{LT}} + \mathcal{L}_{\mathrm{GS}} + \mathcal{L}_{\mathrm{RA}}$ | 32.7 | 28.5 | 16.7 | 45.7 | 74.4 |
| ViT-B/32 | $\mathcal{L}_{\mathrm{GT}}$ (CLIP) | 24.1 | 19.4 | 9.8 | - | - |
| | $\mathcal{L}_{\mathrm{LT}}$ | 25.7(+1.6) | 21.6(+2.2) | 11.3(+1.5) | - | - |
| | $\mathcal{L}_{\mathrm{LT}} + \mathcal{L}_{\mathrm{GS}}$ | 28.8(+4.7) | 24.3(+4.9) | 13.4(+3.6) | - | - |
| | $\mathcal{L}_{\mathrm{LT}} + \mathcal{L}_{\mathrm{GS}} + \mathcal{L}_{\mathrm{RA}}$ | 29.0 | 24.1 | 13.5 | - | - |

**Ablation of Semantically Mismatched Alignment** On the basis of PyramidCLIP paradigm, we further study on the additional effect of $\mathcal{L}_{\mathrm{GT}}$ and $\mathcal{L}_{\mathrm{LS}}$, termed as semantically mismatched alignment, shown in Figure 2(d). The ablation results are listed in Table 9, which reveal that adding semantically mismatched alignment cannot bring stable benefits, even degrades the performance in most cases. Therefore $\mathcal{L}_{\mathrm{GT}}$ and $\mathcal{L}_{\mathrm{LS}}$ are not attached to the PyramidCLIP paradigm, which is consistent with our motivation, *i.e.* addressing the semantics mismatch problem.

**Table 9:** Ablation results of semantics mismatched alignment on the basis of current PyramidCLIP paradigm.

| Image Encoder | Method | ImageNet | MS-COCO | | PASCAL VOC | |
|---|---|---|---|---|---|---|
| | | ZS Top1 | I2T R@1 | T2I R@1 | $AP^{bb}$ | $AP^{bb}_{50}$ |
| ResNet50 | Baseline (PyramidCLIP) | **38.6** | **36.4** | **21.1** | **47.4** | **76.0** |
| | w/ $\mathcal{L}_{\mathrm{GT}}$ | 38.3 | 36.1 | 20.9 | 47.1 | 75.2 |
| | w/ $\mathcal{L}_{\mathrm{LS}}$ | 38.4 | 35.7 | 20.8 | 46.9 | 75.7 |
| | w/ $\mathcal{L}_{\mathrm{GT}} + \mathcal{L}_{\mathrm{LS}}$ | 38.4 | 36.3 | **21.1** | 46.5 | 75.1 |
| ViT-B/32 | Baseline (PyramidCLIP) | **35.9** | 31.7 | **18.8** | - | - |
| | w/ $\mathcal{L}_{\mathrm{GT}}$ | 35.6 | **32.2** | 18.7 | - | - |
| | w/ $\mathcal{L}_{\mathrm{LS}}$ | 35.5 | 29.6 | 17.8 | - | - |
| | w/ $\mathcal{L}_{\mathrm{GT}} + \mathcal{L}_{\mathrm{LS}}$ | 35.5 | 30.9 | 18.1 | - | - |

# G   More Visualizations

In this section, more Grad-CAM heatmaps are visualized through text-to-image retrieval on MS-COCO. We utilize codes provided in [26] to implement Grad-CAM visualization. As shown in Figure 3, CLIP is more likely to focus on the scene or background areas in the images corresponding to the scene descriptions in query texts, while PyramidCLIP pay more attention to salient objects, which benefits from the introduce of cross-level relation alignment. For example, in Figure 3(b), CLIP focuses on areas corresponding to "green field" in the query text, while PyramidCLIP on "horse". In Figure 3(d), CLIP focuses on areas corresponding to "mountain", while PyramidCLIP on "skier with a red jacket". And the same phenomenon can also be seen in Figure 3(a)(c).

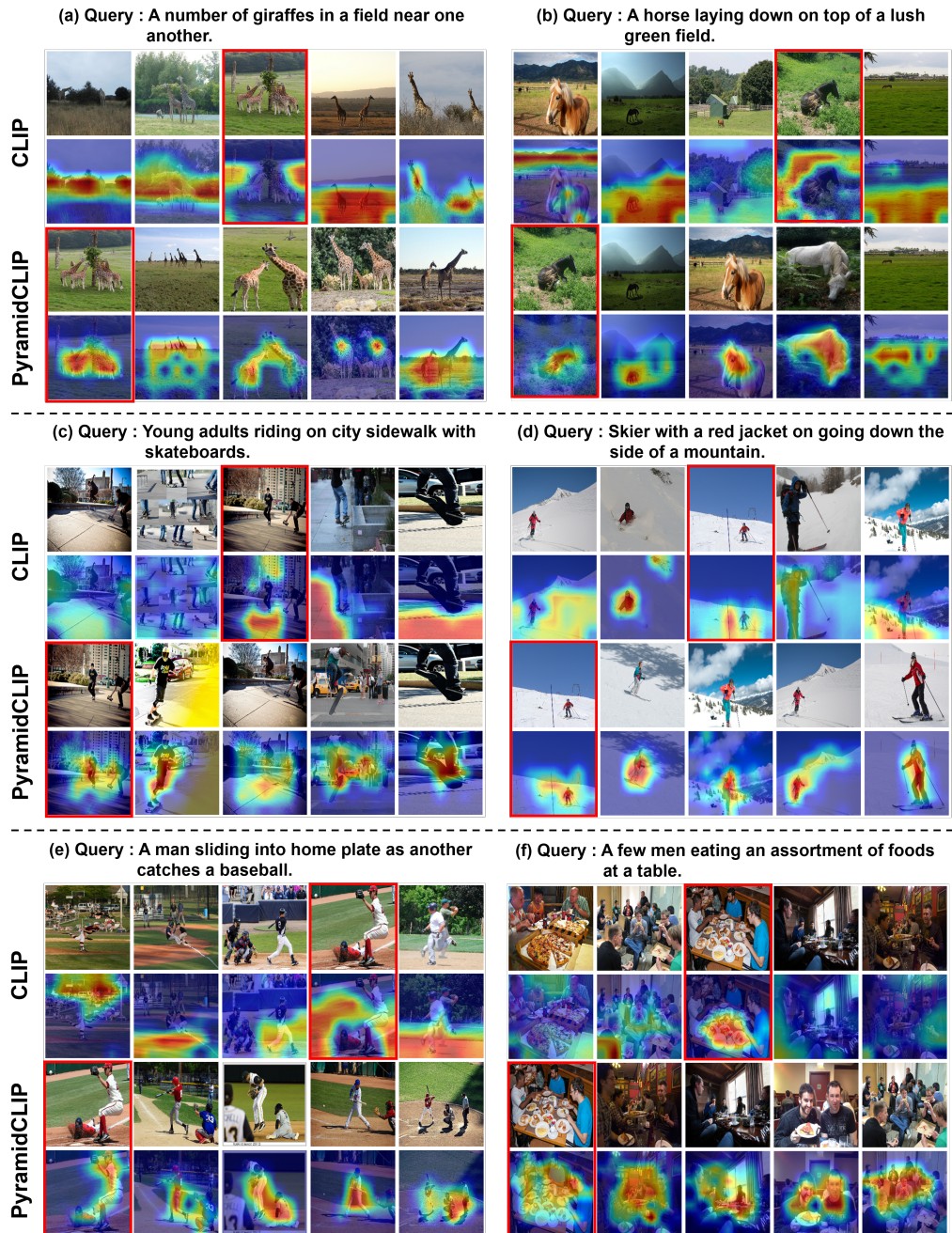

**Figure 3:** More Grad-CAM visualizations through text-to-image retrieval on MS-COCO. From left to right are images from rank1 to rank5. Red box indicates the groundtruth image matched with the query text.