# OpenReview forum: "PyramidCLIP: Hierarchical Feature Alignment for Vision-language Model Pretraining"
_NeurIPS.cc/2022/Conference — NeurIPS 2022 Accept_

### Official Review · Reviewer_QuL5 · 2022-07-05

**Rating:** 7
**Confidence:** 3
**Soundness:** 3 good
**Presentation:** 3 good
**Contribution:** 3 good

**Summary:**

This paper proposes a new framework called PyramidCLIP for vision-language pre-training (VLP). PyramidCLIP first extracts multi-level features from both the visual and linguistic domains and then conducts contrastive learning by aligning visual and linguistic features in both peer-level and cross-level ways. In this way, the vision and language representations learned by PyramidCLIP encode better image-text alignment, which alleviates the semantic mismatch problem that exists in the image-text pairs for pre-training. Moreover, the authors also replace the loss term of the negative samples in contrastive learning with a softened version, to tackle the problem that different image-text pairs may have potential correlation. The empirical results show that PyramidCLIP clearly outperforms the CLIP baseline in a variety of downstream tasks and also achieves SoTA performance on several downstream tasks, as compared with other VLP models.

**Questions:**

* Could you provide an explanation about how the cross-level alignment helps the modelling of relations between salient objects?
* Could provide a more detailed discussion on the difference between PyramidCLIP and the VLP methods that also introduce multi-level semantics?
* Why not compare PyramidCLIP with MVPTR and X-VLM?

**Limitations:**

* The soften objective function treats all the negative samples equally using label smoothing, which is suboptimal considering different image-text pairs should have different degrees of correlation.

**Strengths And Weaknesses:**

### Strengths
* This paper addresses the problem of the quality of image-text pairs (which is an important problem in VLP) using hierarchical feature alignment. The core idea and specific designs of the proposed PyramidCLIP are generally reasonable.
* The authors conduct extensive experiments to demonstrate the effectiveness of PyramidCLIP, covering different backbone architectures (ResNet50 and ViT-B), pre-training datasets of varying sizes and a variety of downstream tasks (zero-shot image classification, zero-shot image-text retrieval, linear probe, object detection and instance segmentation).
* The ablation study is comprehensive, showing that all the components of the proposed method are conducive.
* Qualitative analyses are conducted to intuitively show that PyramidCLIP learns better vision and linguistic representation than CLIP.
* The paper is generally well-written and easy to follow.


### Weaknesses

1. Some specific designs of PyramidCLIP seem counter-intuitive.
* Random crop cannot guarantee the quality, especially for the local view. For example, if the local view is irrelevant to the textual description, minimizing the distance of the corresponding features may confuse the model.
* It is unclear how the cross-level alignment helps the modelling of relations between salient objects. For example, how does the model know that "a table is next to a chair", by contrasting the ROI features to the textual summary or the original text? It is better to provide an intuitive explanation.

2. The difference between PyramidCLIP and existing VLP methods that also introduce multi-level semantics is not clearly discussed. In the related work, the authors state that "Different from methods mentioned above, each level is input to the corresponding encoder individually, without concatenating." Such a difference seems trivial and incremental.

3. Many methods described in the related work are not compared in the experiments, especially the methods that also introduce multi-level semantics (e.g., MVPTR and X-VLM).

---

> ### Author Response · Authors · 2022-08-02
> **Reply to Reviewer QuL5 (Part 2/2)**
>
> **Q3: About why not compare PyramidCLIP with MVPTR and X-VLM?**
>
> The targeted downsteam tasks of PyramidCLIP are quite different from MVPTR/X-VLM (see the table below). The common task is only zero-shot image-text retrieval. However, the model parameters are not comparable since MVPTR/X-VLM includes an additional cross-modal encoder. Besides, the pre-training datasets are also different.  Therefore, comparison with MVPTR/X-VLM does not make much sense. PyramidCLIP is a further extension of CLIP, so it is mainly compared with CLIP-like methods (CLIP, SLIP, DeCLIP, FILIP and DeFILIP) in this paper.
>
> | Method | Targeted Tasks | Model Params | Datasets(Web Crawled) | Datasets(Human Annotated) |
> | --- | :-: | :-: | :-: | :-: |
> | PyramidCLIP |  image classification , image-text retrieval (zero-shot) , object detection and instance segmentation | 100M (RN50) | SBU, CC3M, CC12M, YFCC15M-V1, YFCC15M-V2, LAION99M | None |
> | MVPTR| image-text retrieval (finetune) , VQA, visual entailment, visual grounding  | / | Conceptual Captions (CC) , SBU | MSCOCO, Flirck30k, GQA, OpenImages  |
> | X-VLM | image-text retrieval (zero-shot and finetune) , VQA, NLVR2, visual grounding, image captioning | 216M | SBU, CC3M,  CC12M | COCO,  VG,  Objects365,  OpenImages |

---

> ### Author Response · Authors · 2022-08-02
> **Reply to Reviewer QuL5 (Part 1/2)**
>
> **Q1.1: About Random Crop.**
>
> We assume that the information at the edge of the image usually does not appear in the caption, or is not the main description of the caption. When constructing the local view, the ratio of random crop is 0.5-1, and its expectation is 0.75. Therefore, the discarded area is usually the bounding rectangle, which can eliminate the irrelevant information of the image.
>
> We supplement an experiment in the table below, adjusting the scale of the ramdom crop ratio for the local view from 0.5-1 to using the whole image. It can be seen that the improvement brought by the random crop operation is quite obvious, which further verifies that the random crop operation can indeed discard parts of the image that are not related to the captions.
>
> | Method | YFCC5M |
> | --- | :-: |
> | PyramidCLIP  (random crop=0.5-1) | 31.8 |
> | PyramidCLIP (random crop disabled）| 30.2(-1.6) |
>
> Note that YFCC5M are randomly sampled from YFCC15Mv1.
>
>
> **Q1.2: About Cross-level Relation Alignment.**
>
> Our ROI features are position-sensitive, and its dimension is 2052D, of which 2048D is for appearance, 4D is the position coordinate information. After MHSA aggregates ROI features, the class token contains the relationship between ROIs, both appearance and location. Then we contrast the relation class token with textual summay and the original text. Taking "a table is next to a chair" as an example, we detect the visual feature and position information of "table" and "chair" from the image, and obtain the visual and location relationship of the two after MHSA integration, then contrast the relaiton-containing visual cls token with the caption ("a table is next to a chair"), which can enhance the text encoder's modeling of the relationship between "table" and "chair" and learning of preposition "next".
>
> **Q2: About the difference between PyramidCLIP and existing VLP methods that introduce multi-level semantics.**
>
> Here we mainly discuss the differences between PyramidCLIP and OSCAR/VinVL/MVPTR/X-VLM from five perspectives, that are listed in the table below.
>
> | Method | Paradigm | Encoder Type | Auxiliary Pre-trained Model | Semantics Level | Pre-training Objective |
> | --- | :-: | :-: | :-: | :-: | :-: |
> | OSCAR/VinVL | Single-stream | Transformer-based  | object(-attribute) detector | 2 levels（level concatenation） |  1）Masked Token Loss 2）Contrastive Loss |
> | MVPTR  | Dual-stream + Cross-modal fusion  | Transformer-based | 1) object detector 2) scene graph parser | 2 levels（level concatenation) | 1）Masked Concept Recovering 2）Contrastive Loss 3）Weakly-supervised Phrase Grounding 4）Masked Language Modeling  5）Image Text Matching |
> | X-VLM | Dual-stream + Cross-modal fusion | Transformer-based | None | several levels(each level processed parallelly) | 1）Bounding Box Prediction 2）Contrastive Loss  3）Matching Prediction  4）Masked Language Modeling |
> |PyramidCLIP | Dual-stream | Visual: CNN-based or ViT-based, Text: Transformer-based | 1) object-attribute detector 2）summary extractor | 3 levels(each level processed parallelly) | Contrastive Loss |
>
> It can be seen that PyramidCLIP is a pure dual-stream network that does not need concatenating tokens from two modalities, and only requires contrastive loss for training, which is succinct. Moremore, PyramidCLIP can support more kinds of visual encoders, both CNN and ViT, which is more flexible. In addition, compared to OSCAR/VinVL/MVPTR,  PyramidCLIP can support more granular semantics; compared to X-VLM, PyramidCLIP is more flexible and does not rely on manually annotated bounding boxes and corresponding text descriptions.
>
> Note that X-VLM is similar to PyramidCLIP to some extend regardless of the final cross-modal encoder. However, the several levels of X-VLM only make sense for datasets containing labeled bounding boxes and corresponding annotations like COCO and Visual Genome. For datasets of image-caption pairs, it degenerates into only one level.  Besides, the contrastive loss of X-VLM is employed at each level. Compared with X-VLM, PyramidCLIP also introduces cross-level alignment (contrastive loss) to provide more supervisions in addition to the peer-level alignment at each level. And the three levels of PyramidCLIP make sense for any dataset containing image-caption pairs.

---

> > ### Comment · Reviewer_QuL5 · 2022-08-05
> > **Response to "Reply to Reviewer QuL5 (Part 1/2)"**
> >
> > Thanks for taking the time to respond to my comments.
> >
> > **Q1.1: About Random Crop.**
> >
> > I agree that producing the local view using random crop is an acceptable solution, and, intuitively, the [0.5,1] can indeed cover the relevant regions of the image in most cases. However, the possibility that "the local view is irrelevant to the textual description" is still a potential problem because there is a trade-off in terms of the ratio of the random crop. If the ratio is too small, it is likely that the local view is an irrelevant region. If the ratio is too large, it degenerates into the whole image. Therefore, I think it would be nice to further analyze this trade-off.
> >
> > **Q1.2 and Q2**
> >
> > My concern regarding these two questions is generally addressed. Just one more question: What are the advantages of pure dual-stream models over concatenating tokens from two modalities? Less computation?

---

> > > ### Author Response · Authors · 2022-08-08
> > > **Reply to Reviewer QuL5**
> > >
> > > Q1.1: About Random Crop.
> > >
> > > Here we use salient ROI instances as evaluation metric to quantitatively analyze the quality of Local Crops.
> > >
> > > The ROI instances are salient objects detected by a powerful pre-trained detector. The more ROI instances the Local Crop contains, the more information of the original image the Local Crop contains.
> > >
> > > We define that if area(intersection(ROI, Local Crop))/area(ROI) > 0.5, the ROI is in the Local Crop.
> > >
> > > And the amount of information contained in Local Crop = #(ROIs in local crops)/#(All ROIs), term as InfoiLC.
> > >
> > > In the table below, we list the amount of information contained in the Local Crop (InfoiLC) under different random crop ratios, as well as the corresponding zero-shot ImageNet top-1 accuracy. All are the results on the cc3m dataset.
> > >
> > > | Random Crop Ratio | InfoiLC | IN ZS Top-1 |
> > > | --- | --- | --- |
> > > | 1.0-1.0 | 100% | 23.1 |
> > > | 0.9-1.0 | 99.1% | 22.8 |
> > > | 0.8-1.0 | 97.8% | 23.5 |
> > > | 0.7-1.0 | 96.2% | 23.6 |
> > > | 0.6-1.0 | 94.0% | 24.4 |
> > > | 0.5-1.0 | 93.0% | 24.5 |
> > > | 0.4-1.0 | 90.1% | 24.1 |
> > > | 0.3-1.0 | 87.1% | 24.0 |
> > > | 0.2-1.0 | 80.9% | 23.2 |
> > >
> > > It can be seen that when the the random crop ratio is 0.5-1, the Local Crop still contains 93% of the original image information, and the result is improved by (24.5%-22.8%)=1.7%, compared to the random crop ratio 0.9-1.0 that contains 99.1% of the original image information. We hold the opinion that this improvement is brought by discarding the part of the image that does not match the caption.
> > >
> > > In addition, it can also be seen that when the random crop ratio is too small, the zero-shot ImageNet Top-1 Acc will drop, so 0.5-1.0 is a reasonable value.
> > >
> > > Q1.2 and Q2: About the advantage of pure dual-stream models.
> > >
> > > 1）The parallelization of the dual-stream model is better.
> > >
> > > On the one hand, in practice, for some tasks, such as zero-shot classification, text features can be extracted offline in advance, which is more flexible and friendly in actual deployment.
> > >
> > > On the other hand, on some tasks that require one image and multiple captions to predict, such as zero-shot multi-label classification, dual-stream shows great flexibility. For the dual-stream model, an image and multiple captions are forwarded separately, and then compared at the end, while the single-stream model requires image-text(concatenate) to be forwarded multiple times.
> > >
> > > 2）The computational cost of the dual-stream model is smaller than that of the single-stream, since the number of tokens input to the fusion transformer is the sum of visual tokens and language tokens. (Transformer performs Self-Attention on N tokens, with a complexity of O(N2).)

---

> > > > ### Comment · Reviewer_QuL5 · 2022-08-09
> > > > **Response**
> > > >
> > > > Thank you for the response. My concerns are well-addressed now. It would be nice to see the additional results and discussions in the next version of the paper. I will raise my rating to 7.

---

### Official Review · Reviewer_jqKi · 2022-07-07

**Rating:** 5
**Confidence:** 4
**Soundness:** 2 fair
**Presentation:** 3 good
**Contribution:** 2 fair

**Summary:**

This paper proposes to improve CLIP by adding contrastive losses at different levels. Specifically, for each image in the original data, they construct image views at global, local, and region levels, and also create their corresponding text captions. Therefore, for each image-caption pair in the original dataset, they can create two additional image-caption pairs and the model is trained with additional contrastive losses given the newly constructed data. They also soften the original contrastive loss with a label-smoothing-like technique. Experiments demonstrate improvements over several baselines on zero-shot retrieval, linear probing on image classification,  object detection, and semantic segmentation tasks.

**Questions:**

Would it make more sense to use a keyword extraction model than a summarization model?

**Limitations:**

I am not sure if their method can be scalable as when more data is included, many of the problems mentioned in the paper may be alleviated.

**Strengths And Weaknesses:**

Strengths:
1. The empirical results are strong as they can outperform several popular baselines.
2. The paper is well-structured.
3. I like the idea of softening the contrastive loss objective, which makes sense and the paper shows it works well.

Weaknesses:
1. The comparisons between their model and baselines may not be fair. The paper uses multiple pre-trained models for their model (e.g. text summarization and object detection models) which can introduce more supervision. Also, because their method will create more image-caption pairs for their model, they would have more pretraining data than the baselines.
2. Some designs are questionable. For example, the text captions in image-caption datasets are typically short, whereas they use a pre-trained text summarization model, which is designed for summarizing long documents, to further shorten the captions, which does not make sense to me.
3. While it is true that the pre-training datasets can be noisy, the problem may be alleviated if sufficiently large data are used. It would be good to see if they can keep the performance gain as more data are included. For example, they can plot performance-data size curves for both their model and baselines and see the performance gap when the data size varies.

---

> ### Author Response · Authors · 2022-08-02
> **Reply to Reviewer jqKi (Part 2/2)**
>
> **Q4: About replacing text summarization with keywords.**
>
> We try to use keywords to replace text summarization, but the performance is not as good as text summarization. We analyze it from both qualitative perspective and quantitative perspective.
>
> - Qualitative perspective. Most of the extracted keywords are nouns, and there are relatively few adjectives and prepositions， which will lack some visual patterns and position information between objects. For example, the keyswords for "Soccer player competes for the ball during day of the training camp." is "Soccer, training camp", while the summarization is "Soccer player competes for the ball.", the semantic information of summarization is better.
>
> - Quantitative perspective. We replace the text summarization by keywords and conduct experiments on CC3M. The results are shown in the table below. It can be seen that the results of keywords are not as good as text summarization.
>
> | Method | IN ZS Top-1 Acc |
> | --- | :-: |
> | PyramidCLIP(Summarization) | 24.8 |
> | PyramidCLIP(Keywords) | 24.2(-0.6) |
>
> Note that we use hugging face's open sourced keyword extraction model (https://huggingface.co/yanekyuk/bert-keyword-extractor) to extract keywords, which is finetuned from bert.

---

> > ### Comment · Reviewer_jqKi · 2022-08-05
> > **Reply**
> >
> > Thank you for the response!
> >
> > **Q2&Q4**
> >
> > I'm still not convinced that applying pre-trained text summarization models to image captions is a valid approach. Text summarization datasets typically have quite long inputs (e.g. the average number of tokens of data instances for popular summarization datasets are
> >
> > |  | Reddit| XSum| CNN/DM | WikiHow | NYT | PubMed |
> > | ----- | ----- | -----| ----- | -----| ----- | -----|
> > | Input  |482| 430  | 766 | 580 | 1183 | 444|
> > | Output | 28 | 23 | 58 | 62 | 119 | 210 |
> >
> > As you have shown in this table, the image captions are quite short (even shorter than an output summary of a typical instance) and I cannot imagine how sentences such as 'a street sign sits on top of a stop sign.' (which I randomly sample from COCO) can be summarized.  Therefore, I do not believe applying summarization models to image captions is technically sound. Not including this augmentation method or using keyword extractors (e.g. TextRank) and sentence simplification models would be more plausible to me.
> >
> > **Q1&Q3**
> >
> > Thank you for the additional experiments! I'm not sure if training the models with more data would be more costly than using an object detector and text summarizer to first create instances and then training the model with the augmented data. Also, the performance gap between CLIP and PyramidCLIP indeed seems to narrow as more data are included. However, I think this is a general problem for data augmentation methods and I do not have many concerns in this aspect.

---

> > > ### Author Response · Authors · 2022-08-08
> > > **Reply to Reviewer jqKi**
> > >
> > > Thanks for your prompt reply.
> > >
> > > We would like to stress once more about the soundness of text summarization.
> > >
> > > COCO dataset is a manually annotated dataset with relatively concise captions and has less noise in captions, while the datasets used in large-scale image-text pre-training are usually crawled from the Internet, and the quality of captions is far inferior to that of manually annotated ones.
> > >
> > > We randomly sample one caption and the corresponding text summarization from the several public datasets we used, and list them in the table below.
> > >
> > >
> > > | Dataset | Original Caption | Text Summarization |
> > > | --- | --- | --- |
> > > | CC3M | look an interesting photo , animal is sitting on child | animal is sitting on child.|
> > > | CC12M | A lot of <PERSON> plastic Lego blocks. A lot of <PERSON> plastic Lego blocks stock photography | plastic Lego blocks stock photography.|
> > > | YFCC15M | Alaska cruise day 1, boarding/setting sail: All aboard. 1 Sep 2008: Coral Princess setting sail from Vancouver, Canada. | coral princess set sail from Vancouver, canada. |
> > > | YFCC15MV2 | Hungry Boy Dendrocopos major - Great Spotted Woodpecker - I heard the unmistakalbe peck-peck-peck coming from the top of a pine tree and saw a young woodpecker sitting on a branch Moments later its tired parent landed on the same branch to feed him and then the two of them flew away Not the best picture technically but I liked the moment | a young woodpecker and its tired parent landed on the same branch to feed him |
> > > | LAION99M | Small kid's meal - spaghetti with cherry tomatoes and basil. Colorful italian dinner on white wooden table. Plate captured from above (top view, flat lay). Layout with free copy (text) space. | colorful italian dinner on white wooden table. |
> > >
> > > It can be seen that the captions of these datasets are relatively noisy and contain more redundant information, but after the summarizaion model, captions with relatively compact and concise semantics can be obtained.
> > >
> > > Finally, it is worth mentioning that when we extract the text summarization, if the output length of the summarization model is less than the original caption, we will directly use the original caption, so the length of text summarization must be less than or equal to the caption.

---

> > > > ### Comment · Reviewer_jqKi · 2022-08-09
> > > > **Reply**
> > > >
> > > > Thank you for the response!
> > > >
> > > > The qualitative examples do address my concerns partially. I am still a bit uncomfortable with using summarization models that are pre-trained for summarizing long documents (>400 words) in the news domain in summarizing short captions (10~30 words) in another domain, especially considering that the summarizers can have the hallucination problem. It would be good to include these qualitative examples and discussions in the revised version.
> > > >
> > > > Also, I assume that you mean "if the output length of the summarization model is **greater** than the original caption, we will directly use the original caption"? If the summary length can indeed be greater than the caption length, it confirms my hypothesis that using pre-trained summarizers is not suitable in this setting.
> > > >
> > > > Given the additional examples and considering that this is one of the minor contributions of the paper, I would increase my score by 1, but I do hope the paper can be more explicit about the potential issue of applying summarization models in this setting.

---

> ### Author Response · Authors · 2022-08-02
> **Reply to Reviewer jqKi (Part 1/2)**
>
> **Q1: About the amount of pre-training data.**
>
> We want to emphasize that instead of creating more image-caption pairs, we are mining more information based on existing image-text paris and improving the information utilization of image-caption paris.  Text augmentation and extracting more fine-grained object information with a pre-trained object detector are two very common approaches used in image-text pre-training.  For example, in FILIP,  the text modal uses back-translation operation (please kindly refer to section 3.2 in FILIP[8], https://arxiv.org/pdf/2111.07783v1.pdf).  OSCAL/VinVL/MVPTR all use a pre-trained object detector to extract the features of object, thereby introducing fine-grained information.
>
> At present,  it is far from enough to use the original data for image-text pre-training.  Basically, all mainstream methods will resort to data augmentation, either in visual modality or linguistic modality, so as to improve data utilization, which is reasonable.
>
> **Q2: About text summarization.**
>
> We have counted the average caption length for each dataset we used and the corresponding average summarization length, which is listed in the table below.
>
> 1) It can be seen that the averge text length of some datasets is quite long (which may contain redundance), but the text summarization length is significantly shorter. For example, for YFCC15M dataset, the average length of the original text is 33.8, and the average length of text summarization is 8.5, which is shorted by 75%.
>
> 2) When the original text itself is relatively short, the extracted text summarization still summarize the original caption, but the magnitude is relatively  small.
>
> In conclusion, text summarization works for captions of any length.
>
> | Datasets | Caption Length (avg)  | Summarization Length (avg) |
> | --- | :-: | :-: |
> | CC3M | 10.3 |  10.1 |
> | CC12M | 17.7 | 10.8  |
> | YFCC15M | 33.8 |  8.5 |
> | YFCC15V2 | 16.7 | 8.8  |
> | LAION99M | 9.7 | 8.4|
>
> **Q3.1: About " the noisy problem may be alleviated if sufficiently large data are used".**
>
> 1） Increasing the amount of data may lead to the  improvement of performance, but it will also increase the training time and bring more computation.
>
> 2） With the increase of the data set, the marginal return diminishes, and a large amount of data is required to bring about a siginificant improvement. As shown in the Figure 1(left) of DeCLIP[10] (https://arxiv.org/pdf/2110.05208.pdf), with the pre-training data of CLIP increasing from 15M to 88M, the ImageNet Top-1 accuracy increases by 21%, but with increasing from 88M to 400M, the accuracy only increases by 2.7%. Similar trend can be found in Figure 2 of CLIP[6]. The computational cost of training 400M data is 4.5 times that of 88M, and the cost-performance ratio is relatively low.
>
>
> **Q3.2: About "performance-data size curve."**
>
> In the table below, we record the performance of the baseline and PyramidCLIP with the increase in the amount of data. It can be seen that in the case of a small amount of data, the improvement brought by PyramidCLIP is significant, but when using 83% of the data volume of YFCC15-V1(i.e. 12.5M), PyramidCLIP can still bring a gain of 8.6%, which is considerable.
>
> | Data Volume | CLIP (baseline) | PyramidCLIP |
> | --- | :-: | :-: |
> | 2.5M | 6.9 |  24.8(+17.9) |
> | 5M | 21.7 | 31.8(+10.1)  |
> | 7.5M | 25.1 |  35.4(+10.3) |
> | 10M | 28.2 | 37.3(+9.1)  |
> | 12.5M | 30.6 | 39.2(+8.6) |
>
> The results in the table above are zero-shot ImageNet top-1 accuracy, pre-trained with YFCC15M-V1 subsets of different sizes. Note, the subsets are sampled randomly.
>
> Furthermore,  as shown in the Table 3 of the main text, when we increase the pre-training data volume to 143M, PyramidCLIP can still bring significant improvement compared to the CLIP baseline trained on 143M. Specifically, when the visual encoder is ResNet-50, PyramidCLIP improves zero-shot ImageNet classification by 6.1%, and on the Flickr30K retrieval task, the R@1 of I2T/T2I improves by 5.7%/8% respectively.
>
> Finally, we would like to emphasize that PyramidCLIP solves the common problem existing in vision-language pre-training, that is, the semantic mismatch of image-text and the mutual compatibility between pairs, making PyramidCLIP effective regardless of the size of the pre-training dataset.

---

> ### Author Response · Authors · 2022-08-05
> **Reply to Reviewer jqKi**
>
> Hi, Reviewer  jqKi, did our reply address your concerns? If you have other concerns, please feel free to let us know.

---

### Official Review · Reviewer_WsHx · 2022-07-08

**Rating:** 8
**Confidence:** 4
**Soundness:** 3 good
**Presentation:** 4 excellent
**Contribution:** 4 excellent

**Summary:**

This paper proposes hierarchical feature alignment for vision language pre-training, called PyramidCLIP, which alleviates semantic mismatch as well as mutual compatibility problems, i.e. false positives and false negatives. PyramidCLIP constructs inputs with three levels of semantics in visual and language modalities respectively and then resolves semantic mismatch through peer-level semantic alignment and cross-level relation alignment. In addition, PyramidCLIP adopts a soft form of InfoNCE to deal with mutual compatibility.

**Questions:**

See the weaknesses.

**Limitations:**

Yes

**Strengths And Weaknesses:**

Strengths:

1. This paper is well written and easy to follow.

2. The motivation and solution of the article are clear. More precise hierarchical feature alignment for tackling semantic mismatch, and softening InfoNCE for mutual compatibility.

3. The experiments are well designed, and the results are excellent.


Limitations:

Have the authors adequately addressed the limitations and potential negative social impact of their work? If not, please include constructive suggestions for improvement. Authors should be rewarded rather than punished for being upfront about the limitations of their work and any potential negative societal impact.

1. Needs more explanation about how the training set is constructed.

2. In order to compare with some recently published works(e.g. [1]), it is recommended that the author can supplement the results on smaller scale datasets, such as cc3m.

3. The categories obtained by object detection are simply joined by commas. Whether different joint forms have an impact on the results (e.g. splicing with spaces)?

[1] Robust Cross-Modal Representation Learning with Progressive Self-Distillation. CVPR 2022.

---

> ### Author Response · Authors · 2022-08-02
> **Reply to Reviewer WsHx**
>
> **Q1: About training set.**
>
> Our training set is a collection of publicly available datasets, such as CC3M, YFCC15M, and a subset of 99M data in LAION400M. All these datasets are image-text pairs crawled from the Internet, which may contain a lot of noise.  As for the 99M data seletected from 400M, we use the image-text similarities provided by LAION400M as metric and select the largest 99M. More details can be found in section 4.1 of the main text and A.1 in the supplementary material.
>
> **Q2: Results on CC3M.**
>
> We conduct experiments on CC3M and compare PyramidCLIP with the CLIP baseline, and the results are shown in the table below. For fair comparison, the following experiments are all trained for 32ep with the same hyperparameters, e.g. batch size, learning rate.
>
> | Method | Model Structure | ImageNet Top-1 Acc | ImageNet Top-5 Acc |
> | --- | :-: | :-: | :-: |
> | CLIP | RN50 |  18.9 | 36.3 |
> | PyramidCLIP | RN50 | 27.4 (**+8.5**)  | 47.4 (**+11.1**) |
> | CLIP | ViT-B/32 |  12.4 | 26.4 |
> | PyramidCLIP | ViT-B/32 | 24.0 (**+11.6**)  | 42.5 (**+16.1**) |
>
> Since in PSD (Robust Cross-Modal Representation Learning with Progressive Self-Distillation), the baseline and its method both are trained for a longer time, i.e. 100ep, and the hyperparmaters used are also quite different, we think the comparison is unfair.
>
> **Q3: About different joint forms of categoires.**
>
> We have tried different joint forms, such as splicing with spaces, but they have little effect on the results, and the fluctuation is around 0.2, so we adopt the most intuitive and easy-to-understand joint approach, namely "adj adj adj n, adj adj adj n, ....".

---

### Official Review · Reviewer_BusK · 2022-07-11

**Rating:** 5
**Confidence:** 4
**Soundness:** 3 good
**Presentation:** 3 good
**Contribution:** 3 good

**Summary:**

Under the contrastive CLIP learning framework, the work proposes to utilize more fine-grained information to produce multiple views of both the image and text during training, and hence constructs more contrastive loss terms across different views. During inference/evaluation, only the standard view is used.

Empirically, with 3 different architectures (ResNet50/ViT-B32/ViT-B16), different pretraining data scales and several down-stream datasets, authors show that the proposed approach achieves clear gain over the baseline systems.

**Questions:**

If my understanding above is correct, how would the comparison look like if similar training cost is invested in training the baselines, particularly with soften and LeFF disabled? (NOTE: In my opinion, this is very critical for judging whether the proposed method is really useful. Hence, my final review will largely depend on this.)

**Limitations:**

I don't see any particular problem here.

**Strengths And Weaknesses:**

The work follows a natural motivation and achieves very good empirical gain over some strong baseline systems. Overall, the paper is well written and the empirical study is solid.

A key concern I have is whether the comparison is fair enough. If I understand correctly, for each view of either the image or the text, we need to feed the view into the model once, which leads to roughly 2x - 3x additional computation cost. Again, if my understanding is correct, despite using the same batch size, the actual pretraining cost might be much higher for the proposed method than baselines, making the comparison less information. A better comparison seems to be make the batch size of baselines larger until the training cost is comparable.

---

> ### Author Response · Authors · 2022-08-02
> **Reply to Reviewer BusK**
>
> During the training process, the computation cost of PyramidCLIP is about 2.3 times that of  baseline.  Therefore we compare the baseline (with longer training time and larger batch size) and PyramidCLIP (with soften target and LeFF disabled) and the results are shown in the table below. For fair comparison, all of these experiments are conducted on CC3M, using 16 V100 and the same hyperparameters.
>
> It can be seen that when we adjust the training epoch of the baseline from 8ep to 2times (16ep) or even 3times (24ep), the improvement is  not obvious, and it still underperforms PyramidCLIP trained with 8ep.
>
>
> | Batch Size | Epoch | Model | ImageNet Zero-shot Top1 | Model | ImageNet Zero-shot Top1 |
> | :-: | :-: | :-: | :-: |:-: |:-: |
> | 2048 | 8  | CLIP-ResNet50 | 18.9 | CLIP-ViT-B/32 | 10.6 |
> | 2048 | 16 | CLIP-ResNet50 | 19.7 | CLIP-ViT-B/32  | 11.7 |
> | 2048 | 24 | CLIP-ResNet50 | 20.2 | CLIP-ViT-B/32 | 13.6 |
> | 4096 | 8 | CLIP-ResNet50 | 18.3 | CLIP-ViT-B/32 |  9.7 |
> | 4096 | 16 | CLIP-ResNet50 |  18.9 | CLIP-ViT-B/32 | 10.9 |
> | 2048 | 8 | PyramidCLIP-ResNet50 |  23.8 | PyramidCLIP-ViT-B/32 | 16.8 |
>
> Furthermore, we would like to emphasize the importance of soften target proposed in PyramidCLIP.  As shown in the table above, when we increase the batch size of baseline from 2048 to 4096,  the top-1 acc drops instead. We attribute this phenomenon to that the repetitive rate of text in the existing image-text paired datasets is too high, since they are constructed by crawling images  through a text on the Internet, and it is very likey that one text corresponds to multiple images. (We calculates the text repetitive ratio in the CC3M dataset, which is as high as 30.24%. Same phenomenon also occurs in other pretraining datasets.） Therefore, when the batch size increases, the probability of false negative samples will also increase.
>
> | Batch Size | Epoch | Model | ImageNet Zero-shot Top1 |
> | :-: | :-: | :-: | :-: |
> | 2048 | 8  | PyramidCLIP-ResNet50 (w/o soften target) | 23.8 |
> | 2048 | 8  | PyramidCLIP-ResNet50 (w/ soften target) | 24.5 |
> | 4096 | 8  | PyramidCLIP-ResNet50 (w/o soften target) | 23.4 |
> | 4096 | 8  | PyramidCLIP-ResNet50 (w/ soften target) | 24.3 |
>
> As shown in the table above, soften target can significantly alleviate the problem of false negatives, which brings insight for follow-up research in this field.

---

> ### Author Response · Authors · 2022-08-08
> **Reply to Reviewer BusK**
>
> Hi, Reviewer BusK, did our reply address your concerns? If you have other concerns, please feel free to let us know.

---

### Official Review · Reviewer_e9Ef · 2022-07-12

**Rating:** 6
**Confidence:** 4
**Soundness:** 4 excellent
**Presentation:** 4 excellent
**Contribution:** 4 excellent

**Summary:**

This work introduces to construct an input pyramid with different semantic levels for each modality. It then aligns visual elements and linguistic elements in a hierarchical way. The proposed PyramidCLIP outperforms CLIP with a large margin.



**Questions:**

Please see my comments in the weaknesses part.

**Limitations:**

Yes.

**Strengths And Weaknesses:**

Strength:
- The results are very promising. The model outperforms the sota methods across many datasets.
- The overall framework is easy to follow.

Weakness:
- The idea of introducing ROI features may increase the computational cost significantly.
- In peer-level semantics alignment, the authors introduced coarse-grained global contrast learning and fine-grained local contrast learning. However, I didn’t find the more studies to validate that combining two contrast alignments helps model training.
- The authors described the cross-level relation alignment in Sec 3.3. Three structures are proposed but I suggest the authors providing more analysis to prove their effectiveness.

---

> ### Author Response · Authors · 2022-08-02
> **Reply to Reviewer e9Ef**
>
> **Q1: About the computational cost brought by ROI features.**
>
> 1） The ROI features are extracted offline in advance, so in the training process of all the models, they are extracted once and for all.
>
> 2） Our ROI features are only used in the training phase and do not affect testing and depolyment, i.e., no extra computational burden is introduced in the inference stage, which is parctical.
>
> 3） In the training process, ROI features are input to the latter layers of the visual model (specifically, for ViT, the 9th of the 12 blocks, for ResNet, only the last attention pooling), which requires less computation. Furthermore, Transformer performs Self-Attention on N tokens, with a complexity of O($N^{2}$). For input images of 224 resolution, for ViT-B/16, patch tokens plus cls token, a total of 197 token and  for ResNet-50, the final attention pooling is calculated on 50 tokens. However, we only extract 10 ROI features for each image, so there are only 11 tokens for one image, and the amount of computation introduced is much less than that of images.
>
> We list the FLOPs of the visual and textual model in the following table. Taking ResNet-50 as an example, when the input image is 224x224, the total computational cost of visual model and textual model is 6.26+2.98=9.24GMacs. When the input is ROI features, the computation amount of visual model is very small (0.185G), and the total calculation amount is only 3.165GMacs, which is only 34% of that of 224x224 image input.
>
> | Input | PyramidCLIP-RN50 | PyramidCLIP-ViT-B/16 |
> | --- | :-: | :-: |
> | 224x224 image | 6.26G(V)｜ 2.98G(L) | 17.61G(V) ｜ 2.98G(L)  |
> | ROI features | 0.185G(V) ｜2.98G(L) |  0.229G(V) ｜2.98G(L) |
>
> Note that xG(V) in the above table indicates that the calculation amount in visual modality is XGMacs. L represents linguistical modality.
>
> 4）In order to show that simply increasing the computation amount can not bring significant performance improvement, we increase the number of training epochs of the CLIP baseline to 2 or 3 times. It can be seen that the results of the CLIP baseline trained for a longer time still cannot exceed PyramidCLIP.
>
> | Model | Epoch | IN ZS-shot Top-1 |
> | ---  | :-: | :-: |
> | CLIP-RN50 | 8  | 18.9 |
> | CLIP-RN50 | 16 | 19.7|
> | CLIP-RN50 | 24 | 20.2 |
> | PyramidCLIP-RN50 | 8 | 24.5 |
>
> Note that the above experiments are performed on CC3M, and for fairness, the same hyperparameters are used.
>
> **Q2: About the ablation of peer-level alignment.**
>
> Due to the page limitation of the paper, we put more ablation studies in the supplementary material, including the granular ablation of peer-level alignment. Please refer to Section F.2 (especially Fig 2 and Table 8) in the supplementary material for more details.
>
> **Q3: About the cross-level relation alignment.**
>
> Here, we re-introduce the cross-level relation alignment briefly.
>
> **Motivation:**
>
> &ensp;&ensp; Cross-level relation alignment aims at introducing object-level information, thereby enhancing the modeling of relation between objects, both appearance and position.
>
> **Method:**
>
> 1) First Step (preparation):
>
> PyramidCLIP uses a pre-trained object detector to extract salient object features (termed as ROI features) and corresponding categroy information in the image. ROI features are position-sensitive. In each ROI feature of 2052D, 2048D is for appearance and 4D for position information.
>
>  2) Second Step (encoding):
>
> - Visual Modelling: We input the 2052D ROI feature sequence into one or several transformer blocks, using MHSA to aggregate ROI features. After MHSA integration, the cls token contains the relationship between ROI features, both appearance and position.
>
>    (When the image encoder is ViT, the ROI feature sequence can be directly input to the later layers of the visual encoder for modelling, as shown in Fig 3(b). When the visual model is CNN, we replace the traditional global average pooling with attention pooling and input the ROI feature sequence into the attention pooling for modelling, which is shown in  Fig 3(a).)
>
> - Linguistic Modelling: The category information together with descriptions of all objects are concatenated and input to text encoder to capture the relationship between objects.
>
> 3) Third Step (contrasting):
> - Contrasting visual relation-containing cls token with original caption or text summarization, can promote the text encoder to model the relationship between the nouns and learn the prepositions in the caption.
> - Contrasting textual relation-containing cls token with global view or local view, can make up for the problem that the original caption lacks the description of the salient objects.
>
> The effectiveness of cross-level relation alignment please kindly refer to the Table 5 in the main text.
>
> As for Fig 3(c)，this is Locally-enhanced Feed-Forward(LeFF) we used for ViT, which aims to improve the patch-level local perception and interaction. The effectiveness of LeFF please kindly refer to the last two rows of Table 5 in the main text.

---

### Meta-Review · Area_Chair_oYoM · 2022-08-29

**Recommendation:** Accept
**Confidence:** Certain

**Metareview:**

This paper proposes PyramidCLIP.  It improve contrastive learning method CLIP with more fine-grained information to produce multiple views of both the image and text during training. During inference/evaluation, only the standard view is used. The empirical results with different network architectures at different pretraining data scales show that the proposed Pyramid achieves clear gain over the baseline methods.

The paper is comprehensively discussed, and receives unanimous accept from all reviewers, leading to an ``Accept'' decision. The authors are highly encouraged to revise the paper accordingly. The authors reported results on a customized benchmark, and showed improvement based on its own baseline. In the future, the authors are highly encouraged to report results based on a common benchmark below [*], so that readers can clearly see the position of PyramidCLIP in the context of all other similar papers in the literature.

[*] https://computer-vision-in-the-wild.github.io/eccv-2022/

**Award:**

No

---

### Decision · Program_Chairs · 2022-09-14

Accept